

**Technical note: Applicability of physics-based and machine-learning-based**
**algorithms of geostationary satellite in retrieving the diurnal cycle of cloud base**
**height**
Mengyuan Wang[1], Min Min[1]*, Jun Li[2], Han Lin[3], Yongen Liang[1], Binlong Chen[2],
Zhigang Yao[4], Na Xu[2], Miao Zhang[2]
[1]School of Atmospheric Sciences, Southern Marine Science and Engineering
Guangdong Laboratory (Zhuhai), and Guangdong Province Key Laboratory for
Climate Change and Natural Disaster Studies, Zhuhai 519082, China
[2]Key Laboratory of Radiometric Calibration and Validation for Environmental
Satellites and Innovation Center for FengYun Meteorological Satellite (FYSIC),
National Satellite Meteorological Center (National Center for Space Weather), China
Meteorological Administration, Beijing 100081, China
[3]Key Laboratory of Spatial Data Mining and Information Sharing of Ministry of
Education, National and Local Joint Engineering Research Center of Satellite
Geospatial Information Technology, Fuzhou University, Fuzhou 350108, China
[4]Beijing Institute of Applied Meteorology, Beijing 100029, China
*Correspondence to*: Min Min (minm5@mail.sysu.edu.cn)





**Abstract.** Four distinct retrieval algorithms, comprising two physics-based and two machine-learning (ML) approaches, have been developed to retrieve cloud base height (CBH) and its diurnal cycle from Himawari-8 geostationary satellite observations. Validations have been conducted using the joint CloudSat/CALIOP (Cloud-Aerosol Lidar with Orthogonal Polarization) CBH products in 2017, ensuring independent assessments. Results show that the two ML-based algorithms exhibit markedly superior performance (with a correlation coefficient of R > 0.91 and an absolute bias of approximately 0.8 km) compared to the two physics-based algorithms. However, validations based on CBH data from the ground-based lidar at the Lijiang station in Yunnan province and the cloud radar at the Nanjiao station in Beijing, China, explicitly present contradictory outcomes (R < 0.60). An identifiable issue arises with significant underestimations in the retrieved CBH by both ML-based algorithms, leading to an inability to capture the diurnal cycle characteristics of CBH. The strong consistence observed between CBH derived from ML-based algorithms and the spaceborne active sensor may be attributed to utilizing the same dataset for training and validation, sourced from the CloudSat/CALIOP products. In contrast, the CBH derived from the optimal physics-based algorithm demonstrates the good agreement in diurnal variations of CBH with ground-based lidar/cloud radar observations during the daytime (with an R value of approximately 0.7). Therefore, the findings in this investigation from ground-based observations advocate for the more reliable and adaptable nature of physics-based algorithms in retrieving CBH from geostationary satellite measurements. Nevertheless, under ideal conditions, with an ample dataset of spaceborne cloud profiling radar observations encompassing the entire day for training purposes, the ML-based algorithms may hold promise in still delivering accurate CBH outputs.

**Key words:** Geostationary meteorological satellite; cloud base height; physics-based algorithm; machine learning.

62



## 1 Introduction

Clouds, comprising visible aggregates like atmospheric water droplets, supercooled water droplets, ice crystals, etc., cover roughly 70% of the Earth's surface (Stubenrauch et al., 2013). They play a pivotal role in global climate change, the hydrometeor cycle, aviation safety, and serve as a primary focus in weather forecasting and climate research, particularly storm clouds (Hansen, 2007; Hartmann and Larson, 2002). From advanced geostationary (GEO) and polar-orbiting (LEO, low earth orbit) satellite imagers, various measurable cloud properties, such as fraction, phase, top height, and optical depth, are routinely retrieved. However, the high-quality cloud geometric height (CGH) and CBH, a fundamental macro physical parameter delineating the vertical distribution of clouds, remains relatively understudied and underreported. Nonetheless, for boundary-layer clouds, the cloud base height stands as a critical parameter depending on other cloud-controlling variables. These variables encompass the cloud-base temperature (Zhu et al., 2014), cloud-base vertical velocity (Zheng et al., 2020), activation of CCN (Cloud Condensation Nuclei) at the cloud-base (Rosenfeld et al., 2016; Miller et al., 2023), and the cloud-surface decoupling state (Su et al., 2022). These factors significantly impact convective cloud development and ultimately the climate. Hence, the accurate determination of CBH and its diurnal cycle with high spatial-temporal resolution becomes very important, necessitating comprehensive investigations (Viúdez-Mora et al., 2015; Wang et al., 2020). Such efforts can provide deeper insights into potential ramifications of cloud on radiation equilibrium and global climate systems.

However, as one of the most crucial cloud physical parameters in atmospheric physics, the CBH poses challenges in terms of measurement or estimation from space. Presently, the primary methods for measuring CBH rely on ground-based observations, utilizing tools such as sounding balloons, Mie-scattering lidars, stereo-imaging cloud-height detection technologies, and cloud probe sensors (Forsythe et al., 2000; Hirsch et al., 2011; Seaman et al., 2017; Zhang et al., 2018; Zhou et al., 2019; Zhou et al., 2024). While *in-situ* ground-based observation methods offer highly accurate, reliable, and timely continuous CBH results, they are constrained by localized observation coverage and the sparse distribution of observation sites (Aydin and Singh, 2004). In recent decades, with the rapid





advancement of meteorological satellite observation technology, spaceborne
observing methods have emerged that provide global cloud observations with high
spatio-temporal resolution compared to conventional ground-based remote sensing
methods. In this realm, satellite remote sensing techniques for measuring CBH fall
primarily into two categories: active and passive methods. Advanced active remote
sensing technologies like CloudSat and Cloud-Aerosol Lidar and Infrared Pathfinder
Satellite Observation (CALIPSO) in the National Aeronautics and Space
Administration (NASA) A-Train series can capture global cloud profiles, including
CBH, with high quality by detecting unique return signals from cloud layers using
onboard active millimeter wave radar or lidar. However, their viewing footprints are
limited along the nadir of the orbit, implying that observation coverage remains
confined primarily to a horizontal scale (Min et al., 2022; Lu et al., 2021).
In addition to active remote sensing methods, satellite-based passive remote
sensing technologies can also play an important role in estimating CBH (Meerkötter
and Bugliaro, 2009; Lu et al., 2021). As well known, the physics-based principles and
retrieval methods for cloud top height (CTH) have reached maturity and are now
widely employed in satellite passive remote sensing field (Heidinger and Pavolonis,
2009; Wang et al., 2022). However, the corresponding physical principles or methods
for measuring CBH using satellite passive imager measurements are still not entirely
clear and unified (Heidinger et al., 2019; Min et al., 2020). A recent study by (Yang et
al., 2021) utilized oxygen A-band data observed by the Orbiting Carbon Observatory
2 (OCO-2) to retrieve single-layer marine liquid CBH. Two primary methods are
prominent in retrieving CBH through passive space-based remote sensing techniques.
The first method involves the extrapolation technique for retrieving CBH for clouds
of the same type. For instance, (Wang et al., 2012) proposed a method to extrapolate
CBH from CloudSat using spatial-temporally matched MODIS (Moderate Resolution
Imaging Spectroradiometer) cloud classification data. The second physics-based
retrieval method first approximates the cloud geometric thickness using its optical
thickness. It then employs the previously derived CTH product to compute the
correlated CBH using the respective NOAA (National Oceanic and Atmospheric
Administration) SNPP/VIIRS (Suomi National Polar-orbiting Partnership/Visible
Infrared Imaging Radiometer Suite) products (Noh et al., 2017). Hutchison et al. also
formulated an empirical algorithm that estimates both cloud geometric thickness and
CBH. This algorithm relies on statistical analyses derived from MODIS cloud optical



thickness and cloud liquid water path products (Hutchison et al., 2006; Hutchison,

130  2002).

Machine learning (ML) has proven to be highly effective in addressing nonlinear
problems within remote sensing and meteorology fields, such as precipitation
estimation and CTH retrieval (Min et al., 2020; HåKansson et al., 2018; Kühnlein et
al., 2014). In recent years, several previous studies have leveraged ML-based
algorithms to retrieve CBH, establishing nonlinear connections between CBH and
GEO satellite observations. For instance, Tan et al. (2020) integrated CTH and cloud
optical properties products from Fengyun-4A (FY-4A) GEO satellite with
spatial-temporally matched CBH data from CALIPSO/CloudSat (Tan et al., 2020).
They developed a random forest (RF) model for CBH retrieval. Similarly, Lin et al.
(2022) constructed a gradient boosted regression tree (GBRT) model using U.S.
new-generation Geostationary Operational Environmental Satellites-R Series
(GOES-R) Advanced Baseline Imager (ABI) level 1B radiance data and the ERA5
(the fifth generation ECMWF) reanalysis dataset (Lin et al., 2022). They employed
CALIPSO CBH data as labels to achieve single-layer CBH retrievals. Notably, the
CBH quality of ML-based algorithms was found to surpass that of physics-based
algorithms (Lin et al., 2022). Moreover, Tana et al. (2023) utilized Himawari-8 data
and the random forest algorithm to develop a novel CBH algorithm, achieving a high
correlation coefficient of 0.92 and a low root mean square error (RMSE) of 1.17 km
(Tana et al., 2023).
However, these former studies did not discuss whether both physics-based and
ML-based algorithms of GEO satellite could retrieve the diurnal cycle of CBH well.
This gap in research could be mainly attributed to potential influences from the fixed
LEO satellite (with active radar or lidar) passing time in the previous CBH retrieval
model (Lin et al., 2022). As well known, there are distinct diurnal cycle
characteristics of clouds in different regions across the globe (Li et al., 2022). These
diurnal cycle characteristics primarily stem from the daily solar energy cycle absorbed
by both the atmosphere and Earth's surface. Besides, vertical atmospheric motions are
shaped by imbalances in atmospheric heating and surface configurations, also leading
to a range of cloud movements and structures (Miller et al., 2018). Cloud base plays a
pivotal role in weather and climate processes. It is critical for predicting fog and
cloud-related visibility issues important in aviation and weather forecasting. For
instance, lower cloud bases often lead to more intense rainfall. In climate modeling,



CBH is integral for accurate long-term weather predictions and understanding the
radiative balance of the Earth, which influences global temperatures (Zheng and
Rosenfeld, 2015). Hence, it is crucial to thoroughly investigate the diurnal cycle
features of CBH derived from GEO satellite measurements by comparing them with
ground-based radar and lidar observations (Min and Zhang, 2014; Warren and
Eastman, 2014). In this study, we aim to assess the applicability and feasibility of
both physics-based and ML-based algorithms of GEO satellites in capturing the
diurnal cycle characteristics of CBH.
The subsequent sections of this paper are structured as follows. Section 2
provides a concise overview of the data employed in this study. Following that,
section 3 introduces the four distinct physics/ML-based CBH retrieval algorithms. In
section 4, the CBH results obtained from these four algorithms are analyzed, and
comparisons are drawn with spatially and temporally matched CBHs from
ground-based cloud radar and lidar. Finally, section 5 encapsulates the primary
conclusions and new findings derived from this study.
**2 Data**
In this study, observations from the Himawari-8 (H8) Advanced Himawari
Imager (AHI) are utilized for the retrieval of high spatiotemporal resolution CBH.
Launched successfully by the Japan Meteorological Administration on October 7,
2014, the H8 geostationary satellite is positioned at 140.7°E. The AHI onboard H8
encompasses 16 spectral bands ranging from 0.47 μm to 13.3 μm, featuring spatial
resolutions of 0.5–2 km. This includes 3 visible (VIS) bands at 0.5–1 km, 3
near-infrared (NIR) bands at 1–2 km, and 10 infrared (IR) bands at 2 km. The
H8/AHI can scan a full disk area within 10 minutes, two specific areas within 2.5
minutes, a designated area within 2.5 minutes, and two landmark areas within 0.5
minutes (Iwabuchi et al., 2018). Its enhanced temporal resolution and observation
frequency facilitates the tracking of rapidly changing weather systems, enabling the
accurate determination of quantitative atmospheric parameters (Bessho et al., 2016).
Operational H8/AHI Level-1B data, accessible from July 7, 2015, are freely
available on the satellite product homepage of the Japan Aerospace Exploration
Agency (Letu et al., 2019). The Level-2 cloud products utilized in this study,
including cloud mask (CLM), CTH, cloud effective particle radius (CER), and cloud



optical thickness (COT), are generated by the Fengyun satellite science product
algorithm testbed (FYGAT) (Wang et al., 2019; Min et al., 2017) of the China
Meteorological Administration (CMA) for various applications. It is important to note
that certain crucial preliminary cloud products, such as the cloud mask, have been
validated in prior studies (Wang et al., 2019; Liang et al., 2023). Nevertheless, before
initiating CBH retrieval, it is imperative to validate the H8/AHI cloud optical and
microphysical products from the FYGAT retrieval system. This validation is carried
out by using analogous MODIS Level-2 cloud products as a reference. Additional
details regarding the validation of cloud products are provided in the Appendix A
section.

In addition to the H8/AHI Level-1/2 data, the Global Forecast System (GFS)
numerical weather prediction (NWP) data are employed for CBH retrieval in this
study. The variables include land/sea surface temperature and the vertical profiles of
temperature, humidity, and pressure. Operated by the U.S. NOAA (Kalnay et al.,
1996), the GFS serves as a global and advanced NWP system. The operational GFS
system routinely delivers globally high-quality and gridded NWP data at 3-hour
intervals, with four different initial forecast times per day (00:00, 06:00, 12:00, and
18:00 UTC). The three-dimensional NWP data cover the Earth in a 0.5°×0.5° grid
interval and resolve the atmosphere with 26 vertical levels from the surface (1000 hPa)
up to the top of the atmosphere (10 hPa).

As previously mentioned, the official MODIS Collection-6.1 Level-2 cloud
product Climate Data Records are utilized in this study to validate the H8/AHI cloud
products (CTH, CER, and COT) generated by the FYGAT system. MODIS sensors
are onboard NASA Terra and Aqua polar-orbiting satellites. Terra functions as the
morning satellite, passing through the equator from north to south at approximately
10:30 local time, while Aqua serves as the afternoon satellite, traversing the equator
from south to north at around 13:30 local time. As a successor to the NOAA
Advanced Very High Resolution Radiometer (AVHRR), MODIS features 36
independent spectral bands and a broad spectral range from 0.4 μm (VIS) to 14.4 μm
(IR), with a scanning width of 2330 km and spatial resolutions ranging from 0.25 to
1.0 km. Recent studies (Baum et al., 2012; Platnick et al., 2017) have highlighted
significant improvements and collective changes in cloud top, optical, and
microphysical properties from Collection-5 to Collection-6.



In addition to the passive spaceborne imaging sensors mentioned above, the
CloudSat satellite, equipped with a 94-GHz active cloud profiling radar (CPR), holds
the distinction of being the first sun-synchronous orbit satellite specifically designed
to observe global cloud vertical structures and properties. It is part of the A-Train
(Afternoon-Train) series of satellites, akin to the Aqua satellite, launched and
operated by NASA (Heymsfield et al., 2008). CALIPSO is another polar-orbiting
satellite within the A-Train constellation, sharing an orbit with CloudSat and trailing
it by a mere 10–15 seconds. CALIPSO is the first satellite equipped with an active
dual-channel CALIOP at 532 and 1064 nm bands (Hunt et al., 2009). Both CloudSat
and CALIPSO possess notable advantages over passive spaceborne sensors due to the
94-GHz radar of CloudSat and the joint return signals of lidar and radar on CALIPSO.
These features enhance their sensitivity to optically thin cloud layers and ensure
strong penetration capability, resulting in more accurate CTH and CBH detections
compared to passive spaceborne sensors (CAL_LID_L2_05kmCLay-Standard-V4-10).
The joint cloud type products of 2B-CLDCLASS-LIDAR, derived from both
CloudSat and CALIPSO measurements, offer a comprehensive description of cloud
vertical structure characteristics, cloud type, CTH, CBH, etc. The time interval
between each profile in this product is approximately 3.1 seconds, and the horizontal
resolution is 2.5 km (along track)×1.4 km (cross-track). Each profile is divided into
125 layers with a 240-m vertical interval. For more details on
2B-CLDCLASS-LIDAR products, please refer to the CloudSat official product
manual (Sassen and Wang, 2008). Please note that for this study, we utilized one-year
H8/AHI data and matched it with the joint CloudSat/CALIOP data from January 1 to
December 31 of 2017.
**3 Physics/machine-learning based cloud-base height algorithms**
**3.1 GEO Cloud-base height retrieval algorithm from the interface data processing**
**segment of the Visible Infrared Imaging Radiometer Suite**
The Joint Polar Satellite System (JPSS) program is a collaborative effort between
NASA and NOAA. The operational CBH retrieval algorithm, part of the 30
Environmental Data Records (EDR) of JPSS, can be implemented operationally
through the Interface Data Processing Segment (IDPS) (Baker, 2011). In this study,
our geostatic satellite CBH retrieval algorithm aligns with the IDPS CBH




algorithm developed by (Baker, 2011). Utilizing the geostationary H8/AHI cloud
products discussed earlier, this new GEO CBH retrieval algorithm is succinctly
outlined below.

The new GEO IDPS CBH algorithm initiates the process by first retrieving the
cloud geometric thickness (CGT) from bottom to top. Subsequently, CGT is
subtracted from the corresponding cloud top height (CTH) to calculate CBH (CBH =
CTH − CGT). The algorithm is divided into two independent executable modules
based on cloud phase, distinguishing between liquid water and ice clouds. CBH of
water cloud retrieval requires Cloud Optical Thickness (COT or $D_{COT}$) and Effective
Radius (CER or $R_{eff}$) as inputs. For ice clouds, an empirical equation is employed for
CBH retrieval. However, the standard deviations of error in IDPS CBH for individual
granules often exceed the JPSS VIIRS minimum uncertainty requirement of ±2km
(Noh et al., 2017). The accuracy of IDPS algorithm-derived CBHs can be directly
affected by several factors, including cloud optical thickness, cloud effective particle
size, the presence of multiple-layered cloud systems, lack of solar illumination, and
highly reflective surfaces such as snow or ice surfaces. For a more comprehensive
understanding of this CBH algorithm, please refer to the IDPS algorithm
documentation (Baker, 2011).

**3.2 GEO Cloud-base height retrieval algorithm implemented in the Clouds from
Advanced Very High Resolution Radiometer Extended system**

As mentioned above, the accuracy of the GEO IDPS algorithm is highly
dependent on the initial input parameters such as cloud phase, $D_{COT}$ and $R_{eff}$, which
may introduce some uncertainties in the final retrieval results. In contrast, a more
reliable statistically-based algorithm is proposed and implemented here, which is
named the GEO CLAVR-x (Clouds from AVHRR Extended, NOAA's operational
cloud processing system for the AVHRR) CBH algorithm, and it mainly refers to
NOAA AWG CBH algorithm (ACBA). Previous studies have also demonstrated a
correlation coefficient of 0.569 and a root mean square error (RMSE) of 2.3 km for
the JPSS VIIRS CLAVR-x CBH algorithm. It is anticipated that this algorithm will
also be employed for the NOAA GORS-R geostationary satellite imager (Noh et al.,
2017; Seaman et al., 2017).

Similar to the GEO IDPS CBH retrieval algorithm mentioned earlier, the GEO
CLAVR-x CBH retrieval algorithm also initially obtains CGT and CTH, subsequently

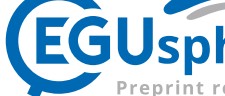

calculating CBH by subtracting CGT from CTH (CTH−CGT). However, the specific
calculation method for the CGT value differs. This algorithm is suitable for both
single-layer and multi-layer clouds, computing CBH using the CTH at the top layer of
the cloud. In comparison with the former GEO IDPS CBH algorithm, the GEO
CLAVR-x CBH algorithm considers two additional cloud types: deep convection
clouds and thin cirrus clouds. For more details on this CLAVR-x CBH algorithm,
please refer to the original algorithm documentation (Noh et al., 2017).

**3.3 Random-forest-based cloud-base height estimation algorithm**

RF, one of the most significant ML algorithms, was initially proposed and
developed by (Breiman, 2001). It is widely employed to address classification and
regression problems based on the law of large numbers. The law of large numbers
states that when independent and identically distributed random experiments are
repeatedly conducted, the average of the results will converge to the expected value as
the number of trials increases. In RF algorithms, it primarily serves to increase
randomness and independence in model construction, thus enhancing the model's
stability and generalizability. Here, the RF method utilizes a forest of trees, serving as
an integrated algorithm that enhances overall model accuracy and generalization by
combining multiple weak classifiers. The final prediction is calculated through voting
or averaging. The RF method is well-suited for capturing complex or nonlinear
relationships between predictors and predictands. As mentioned earlier, this statistical
or ML-based algorithm has been already proven successful in retrieving CTH and
CBH (Min et al., 2020; Tan et al., 2020).
In this study, two distinct ML-based GEO CBH algorithms, namely VIS+IR and
IR-single (only uses observations of H8/AHI IR channels), are devised to retrieve or
predict the CBH using different sets of predictors. The RF training of the chosen
predictors is formulated as follows:
$CBH = RF_{reg}[x_1, x_2, …, x_n]$,                                          (1)
where $RF_{reg}$ denotes the regression Random Forests model, and $x_i$ represents the $i$th
predictor. The selected predictors from H8/AHI for both the VIS+IR and IR RF
model training and prediction are detailed in Table 1, mainly referencing Min et al.
(2020) and Tan et al. (2020). The VIS+IR algorithm retrieves CBH based on NWP
data (atmospheric temperature and altitude profiles, total precipitable water (TPW),
surface temperature), surface elevation, air mass 1 (air mass 1=1/cos(view zenith



angle)), and air mass 2 (air mass 2=1/cos(solar zenith angle)). The rationale for
choosing air mass and TPW is their ability to account for the potential absorption
effect of water vapor along the satellite viewing angle. The predictors in CBH
retrieval also include the IR band Brightness Temperature (BT) and VIS band
reflectance. The IR-single algorithm selects the same Global Forecast System (GFS)
NWP data as the VIS+IR algorithm but employs different view zenith angles and
azimuth angles.

To optimize the RF prediction model, the hyperparameters of the RF model are
tuned individually. The parameters and their dynamic ranges involved in tuning the
RF prediction models include the number of trees [100, 200, 300, 400, 500], the
maximum depth of trees [10, 20, 30, 40, 50], the minimum number of samples
required to split an internal node [2, 4, 6, 8, 10], and the minimum number of samples
required to be at a leaf node [1, 3, 5, 7, 9]. In this study, we set the smallest number of
trees in the forest to 100 and the maximum depth of the tree to 40.

The performance of RF models will be assessed using mean absolute error
(MAE), mean bias error (MBE), root mean square error (RMSE), correlation
coefficient (R), and standard deviation (STD) scores based on the testing dataset.
These scores mentioned above are used to understand different aspects of the
predictive performance of model: MAE and RMSE provide insights into the average
error magnitude, MBE indicates bias in the predictions, R evaluates the linear
association between observed and predicted values, and STD assesses the variability
of the predictions. In the RF IR-single algorithm, 581,783 matching points are
selected from H8/AHI and CloudSat data for 2017. Seventy percent of these points
are randomly assigned to the training dataset, and the remainder serves as the testing
dataset. For the RF VIS+IR algorithm, a total of 418,241 matching points are chosen,
with 70% randomly allocated to the training set. It's important to note that the two
training datasets in CloudSat will also be used to verify the CBHs obtained by cloud
radar and lidar. The statistical formulas for evaluation are as follows:
$\text{MAE} = \frac{1}{n}\sum_{i=1}^{n}|y_i - x_i|,$ (2)
$\text{MBE} = \frac{1}{n}\sum_{i=1}^{n}(y_i - x_i),$ (3)
$\text{RMSE} = \sqrt{\frac{1}{n}\sum_{i=1}^{n}(y_i - x_i)^2},$ (4)



$$R = \frac{\sum_{i=1}^{n}(y_i-\bar{y})(x_i-\bar{x})}{\sqrt{\sum_{i=1}^{n}(y_i-\bar{y})^2}\sqrt{\sum_{i=1}^{n}(x_i-\bar{x})^2}}, \tag{5}$$

$$STD = \sqrt{\frac{1}{n-1}\sum_{i=1}^{n}(x_i - \bar{x})^2}, \tag{6}$$

where $n$ is the sample number, $y_i$ is the $i$th CBH retrieval result, and $x_i$ is the $i$th joint CloudSat/CALIOP CBH product.

## 4 Results and Discussions

### 4.1 Comparisons with the joint CloudSat/CALIPSO cloud-base height product

The H8/AHI satellite CBH data retrieved by the four algorithms are matched spatially and temporally with the 2B-CLDCLASS-LIDAR cloud product from joint CloudSat/CALIPSO observations in 2017. Fig. 1 displays a comparison of CBH results over the full disk at 02:00 UTC on January 1, 2017, retrieved by the GEO IDPS algorithm, the GEO CLAVR-x algorithm, the RF VIS+IR algorithm, and the RF IR-single algorithm. A similar distribution pattern and magnitude of CBHs retrieved by these four independent algorithms can be observed in Fig. 1. However, notable differences exist between physics-based and ML-based algorithms. Further comparisons are conducted and analyzed with spaceborne and ground-based lidar and radar observations in the subsequent sections of this study.

Fig. 2 presents the density scatter plot of the CBHs retrieved from the GEO IDPS and GEO CLAVR-x algorithms compared with the CBHs from the joint CloudSat/CALIPSO product, along with the related scores of MAE, MBE, RMSE, and R calculated and labeled in each panel. The calculated R exceeds the 95% significance level ($p < 0.05$). For the GEO IDPS algorithm, the R is 0.62, the MAE is 1.826 km, and the MBE and RMSE are -0.232 and 2.642 km (Fig. 2a). In comparison, (Seaman et al., 2017) compared the operational VIIRS CBH product retrieved by the similar SNPP/VIIRS IDPS algorithm with the CloudSat CBH results. In their results, the R is 0.569, and the RMSE is 2.3 km. For the new GEO CLAVR-x algorithm (Fig. 2b), the R is 0.647, and the RMSE is 2.91 km. The larger RMSE from two independent physics-based CBH algorithms demonstrate a slightly poorer performance and precision of these retrieval algorithms for GEO satellites. Particularly, the larger RMSEs (2.642 and 2.91 km) indicate weaker stabilities of the GEO IDPS and CLAVR-x CBH algorithms. In this figure, more samples can be found



near the 1:1 line, implying the good quality of retrieved CBHs. However, in stark
contrast, quite a number of CBH samples retrieved by both GEO IDPS and GEO
CLAVR-x algorithms (compared with the official VIIRS CBH product) fall below 1.0
km, indicating relatively large errors when compared with the joint
CloudSat/CALIPSO CBH product. The poor predictive performance of physics-based
algorithm for samples with a CBH lower than 1 km is likely due to insufficient cloud
base information in the visible band observation data. Moreover, Fig. 2 reveals that
relatively large errors are also found in the CBHs lower than 2 km for the four
independent algorithms, primarily caused by the weak penetration ability of VIS or IR
bands on thick and low clouds.
Referring to the joint CloudSat/CALIPSO CBH product, Fig. 2c and 2d present
the validations of the CBH results retrieved from two ML-based algorithms using the
VIS+IR (only retrieving the CBH during the daytime) and IR-single models. Fig. 2c
demonstrates better consistency of CBH between the VIS+IR model and the joint
CloudSat/CALIPSO product with R = 0.905, MAE = 0.817 km, MBE = 0.425 km,
and RMSE = 1.706 km. Fig. 2d also displays a relatively high R of 0.876 when
validating the IR-single model, with MAE = 0.882, MBE = −0.445, and RMSE =
1.995. Therefore, both VIS+IR and IR-single models can obtain high-quality CBH
retrieval results from geostationary imager measurements. In comparison, previous
studies also proposed similar ML-based algorithms for estimating CBH using FY-4A
satellite imager data. For example, (Tan et al., 2020) used the variables of CTH, $D_{\mathrm{COT}}$,
$R_{\mathrm{eff}}$, cloud water path, longitude/latitude from FY-4A imager data to build the training
and prediction model and obtained CBH with MAE=1.29 km and R=0.80. In this
study, except CTH, the other Level-2 products and geolocation data
(longitude/latitude) used in (Tan et al., 2020) are abandoned, while the matched
atmospheric profile products (such as temperature and relative humidity) from NWP
data are added. These changes in ML-based model training and prediction lead to
more accurate CBH retrieval results. Note that, in accordance with the previous study
conducted by (Noh et al., 2017), we excluded CBH samples obtained from
CloudSat/CALIPSO that were smaller than 1 km in our comparisons. This exclusion
was primarily due to the presence of ground clutter contamination in the CloudSat
CPR data (Noh et al., 2017).
Fig. 3 displays two cross-sections of CBH from various sources overlaid with
CloudSat radar reflectivity (unit: dBZ) for spatially and temporally matched cases.



The periods covered are from 03:16 to 04:55 UTC on January 13, 2017 (154.0°E–
160.0°E; 40.56°S–53.39°S) and from 05:38 to 07:17 UTC on January 14, 2017
(107.1°E–107.8°E; 8.35°N–11.57°N). The CloudSat radar reflectivity and joint
CloudSat/CALIPSO product provide insights into the vertical structure or distribution
of clouds and their corresponding CBHs. The results from the four GEO CBH
retrieval algorithms (GEO IDPS, GEO CLAVR-x, RF VIS+IR model, and RF
IR-single model) mentioned earlier are individually marked with different markers in
each panel. According to Fig. 3a, the GEO IDPS algorithm faces challenges in
accurately retrieving CBHs for geometrically thicker cloud samples near 157°E.
Optically thick mid- and upper-level cloud layers may obscure lower-level cloud
layers. However, the CBH results retrieved by the GEO IDPS algorithm near 155°E
(in Fig. 3a) and 107.4°E (in Fig. 3b) align with the joint CloudSat/CALIPSO CBH
product. It is worth noting that the inconsistency observed between 107.2°E and
107.3°E in Fig. 3b, specifically regarding the CBHs around 1 km obtained from
CloudSat/CALIPSO, can likely be attributed to ground clutter contamination in the
CloudSat CPR data (Noh et al., 2017). The GEO CLAVR-x algorithm achieves
improved CBH results compared to the GEO IDPS algorithm. It can even retrieve
CBHs for some thick cloud samples that are invalid when using the GEO IDPS
algorithm. However, the CBHs from the GEO CLAVR-x algorithm are noticeably
higher than those from the joint CloudSat/CALIPSO product. In contrast, the CBHs
from the two ML-based algorithms show substantially better results than those from
the other two physics-based algorithms. Particularly, the ML-based VIS+IR model
algorithm yields the best CBH results. However, compared with those from the two
physics-based algorithms, the CBHs from the two ML-based algorithms still exhibit a
significant error around 5 km.
Since the two RF models (VIS+IR and IR-single) select 230 typical variables to
fit CBHs, the important scores of these predictors in the two ML-based algorithms are
ranked for better optimization. In the VIS+IR model, the top-ranked predictors are
CTH and CTT from the H8/AHI Level-2 product (see Fig. B1 in Appendix B). It's
important to note that $D_{COT}$ is a crucial and sensitive factor for these ML-based
algorithms. Retrieving CBH samples with relatively low $D_{COT}$ remains challenging
due to the low signal-to-noise ratio when $D_{COT}$ is low (Lin et al., 2022). To address
this issue, samples with $D_{COT}$ less than 1.6 are filtered in the VIS+IR model, and
samples with relatively large BTs at Channel-14 are filtered in the IR-single model.





This filtering process significantly improves the R value from 0.869 to 0.922 in the
VIS+IR model and from 0.868 to 0.911 in the IR-single model. For more details on
the algorithm optimization, please refer to Appendix B.

**4.2 Comparisons with the ground-based lidar and cloud radar measurements**

Lidar actively emits lasers in different spectral bands into the air. When the laser
signal encounters cloud particles during transmission, a highly noticeable
backscattered signal is generated and received (Omar et al., 2009). When lidar
measures clouds, the intensity of the echo signal from the cloud to the laser satisfies
the lidar equation as follows:
$$P(r) = C * \beta(r) * r^{-2} * exp\left[-2\int_0^r \sigma(z)dz\right], \tag{7}$$
where $P$ (r) is the intensity of the atmospheric backscattered signal received by the
laser telescope from the emission point in distance $r$ (unit: Watt or W); $C$ is the lidar
system instrumentation constant (unit: W·km$^3$·sr); $r$ is the detection distance (unit:
km); $\beta$(r) is the backscattering coefficient at the emission point in distance $r$ (unit:
km$^{-1}$·sr$^{-1}$); $\sigma$(z) is the extinction coefficient at the distance emission point in distance
$z$ (unit: km$^{-1}$). This return signal is markedly distinct from atmospheric aerosol
scattering signals and noise, making CBH easily obtainable from the signal difference
or mutation (Sharma et al., 2016). In this study, continuous ground-based lidar data
from the Twin Astronomy Manor in Lijiang City, Yunnan Province, China (26.454°N,
100.0233°E, altitude = 3175 m) are used to evaluate the diurnal cycle characteristics
of CBHs retrieved using GEO satellite algorithms (Young and Vaughan, 2009). The
geographical location and photo of this station are shown in Fig. 4.
The ground-based lidar data at Lijiang station on December 6, 2018, and January
8, 2019, are selected for validation. The number of available and spatially-temporally
matched CBH sample points from ground-based lidar is 78 and 64 on December 6,
2018, and January 8, 2019, respectively. Fig 5a and 5b show the point-to-point CBH
comparisons between ground-based lidar and four GEO satellite CBH algorithms on
December 6, 2018, and January 8, 2019. It is worth noting that the retrieved CBHs of
the two physics-based algorithms on December 6, 2018, are in good agreement with
the reference values from the lidar measurements, and, in particular, the GEO
CLAVR-x algorithm can obtain better results. From the results on January 8, 2019,
more accurate diurnal cycle characteristics of CBHs are revealed by the GEO



CLAVR-x algorithm than by the GEO IDPS algorithm.
Compared with the CBHs measured by ground-based lidar, the statistics of the
results retrieved from the GEO IDPS algorithm are R = 0.67, MAE = 3.093 km, MBE
= 0.856 km, and RMSE = 3.609 km (Fig. 5c). However, for cloud samples with CBH
below 7.5 km, the GEO IDPS algorithm shows an obvious underestimation of CBH in
Fig. 5c. For the GEO CLAVR-x algorithm, it can also be seen that the matched
samples mostly lie near the 1:1 line with R = 0.773 (the optimal CBH algorithm),
MAE = 1.319 km, MBE = 0.222 km, and RMSE = 1.598 km. In addition, this figure
also shows the CBH comparisons between the ML-based VIS+IR model/IR-single
model algorithms and the lidar measurements, revealing that the retrieved CBH
results from the ML-based VIS+IR model are better than those from the ML-based
IR-single model algorithm. The comparison results between the CBHs of the
ML-based VIS+IR model algorithm and the lidar measurements are around the 1:1
line, with smaller errors and R = 0.599. In contrast, the R between the CBHs of the
ML-based IR-single model algorithm and the lidar measurements is only 0.494, with a
relatively large error. By comparing the retrieved CBHs with the lidar measurements
at Lijiang station, it indicates that CBH results from two physics-based algorithms are
remarkably more accurate, particularly that the GEO CLAVR-x algorithm can well
capture diurnal variation of CBH.
To further assess the accuracy and quality of the diurnal cycle of CBHs retrieved
with these algorithms, CBHs from another ground-based cloud radar dataset covering
the entire year of 2017 are also collected and used in this study. The observational
instrument is a Ka-band (35 GHz) Doppler millimeter-wave cloud radar (MMCR)
located at the Beijing Nanjiao Weather Observatory (a typical urban observation site)
(39.81°N, 116.47°E, altitude = 32 m; see Fig. 4), performing continuous and routine
observations. The MMCR provides a specific vertical resolution of 30 m and a
temporal resolution of 1 minute for single profile detection, based on the radar
reflectivity factor. In a previous study (Zhou et al., 2019), products retrieved by this
MMCR were utilized to investigate the diurnal variations of CTH and CBH, and
comparisons were made between MMCR-derived CBHs and those derived from a
Vaisala CL51 ceilometer. The former study also found that the average correlation
coefficient (R) of CBHs from different instruments reached up to 0.65. It is worth
noting that the basic physics principle for detecting cloud base height from both
spaceborne cloud profiling radar and ground-based cloud radar and lidar



measurements is the same. All these algorithms of detecting CBH based on the
manifest change of return signals between CBH and the clear sky atmosphere in the
vertical direction (Huo et al., 2019; Ceccaldi et al., 2013). The joint spaceborne
CloudSat/CALIPSO detection might face limitations in penetrating extremely dense,
optically thick, or areas with heavy precipitation clouds. Hence, in comparison, the
CBH values gathered from ground-based lidar and cloud radar measurements are
expected to be more accurate than the data derived from spaceborne
CloudSat/CALIPSO detection.
Similar to Fig. 5, Fig. 6 presents two sample groups of CBH results from the
cloud radar at Beijing Nanjiao station relative to the matched CBHs from the four
retrieval algorithms (GEO IDPS, GEO CLAVR-x, ML-based IR-single, ML-based
VIS+IR) on April 9–10 and July 26–28, 2017. Due to the density of points in the
one-year time series, the point-to-point CBH comparison results for the entire year are
not displayed here (monthly results are shown in the supplementary document).
Similar to the results at Lijiang station discussed in Fig. 5, we observe better and more
robust performances in retrieving diurnal cycle characteristics of CBH from the two
physics-based CBH retrieval algorithms. In contrast, more underestimated CBH
samples are retrieved by the two ML-based algorithms.
To further investigate the diurnal cycle characteristics of retrieved CBH from
GEO satellite imager measurements, Fig. 7 presents box plots of the hourly CBH
errors (relative to the results of cloud radar at Beijing Nanjiao station) in 2017 from
the four different CBH retrieval algorithms. Remarkably, there are significant
underestimations of the CBHs retrieved from the two ML-based algorithms. The
ML-based VIS+IR method achieves relatively better results than the ML-based
IR-single method during the daytime. Comparing the two ML-based algorithms, the
errors of the IR-single model algorithm have a similar standard deviation (2.80 km) to
those of the VIS+IR model algorithm (2.69 km) during the daytime. For the IR-single
model algorithm, it can be applied during both daytime and nighttime, its nighttime
performance degrades slightly, with an averaged RMSE (3.88 km) higher than that of
daytime (3.56 km). To the best of our knowledge, there is no alternative nighttime
CBH product for geostationary satellite imagers right now. The nighttime CBH of the
IR-single model algorithm is the only choice that should be used with discretion.
Fig. 8 shows the comparisons of hourly MAE, MBE, RMSE, and R relative to the
CBHs from the cloud radar at Beijing Nanjiao station during daytime between four





retrieval algorithms in 2017. The RMSE of the two ML-based algorithms shows
stable diurnal variation. It is noted that all algorithms have lower correlation
coefficients (R) at sunrise, around 07:00 local time, which improve as the day
progresses. However, the GEO CLAVR-x algorithm stands out for its relatively higher
and more stable in R and RMSE during daytime.
Fig. 9a displays scatter plots and relevant statistics of the CBHs retrieved from
the GEO IDPS algorithm against the CBHs from cloud radar. The CBHs from the
GEO IDPS algorithm align well with the matched CBHs from cloud radar at Beijing
Nanjiao station, with R = 0.515, MAE = 2.078 km, MBE = 1.168 km, and RMSE =
2.669 km. In Fig. 9b, the GEO CLAVR-x algorithm shows better results with R =
0.573, MAE = 2.059 km, MBE = −0.204 km, and RMSE = 2.601 km. It is not
surprising that Figs. 8c and 8d reveal obvious underestimated CBH results from the
two ML-based CBH algorithms. Particularly, the CBH results from the ML-based
VIS+IR model algorithm concentrate in the range of 2.5 km to 5 km. Therefore, Fig. 5
to Fig. 9 further substantiates the weak diurnal variations captured by ML-based
techniques, primarily attributed to the scarcity of comprehensive CBH training
samples throughout the entire day. Besides, although the two robust physics-based
algorithms of GEO IDPS and GEO CLAVR-x (the optimal one) can retrieve
high-quality CBHs from H8/AHI data, especially the diurnal cycle of CBH during the
daytime, they still struggle to retrieve CBHs below 1 km.
**5. Conclusions and discussion**
To explore and argue the optimal and most robust CBH retrieval algorithm from
geostationary satellite imager measurements, particularly focusing on capturing the
typical diurnal cycle characteristics of CBH, this study employs four different
retrieval algorithms (two physics-based and two ML-based algorithms). High
spatial-temporal resolution CBHs are retrieved using the H8/AHI data from 2017 and
2018. To assess the accuracies of the retrieved CBHs, point-to-point validations are
conducted based on spatially-temporally matched CBHs from the joint
CloudSat/CALIOP product, as well as ground-based lidar and cloud radar
observations in China. The main findings and conclusions are outlined below.
Four independent CBH retrieval algorithms, namely physics-based GEO IDPS,
GEO CLAVR-x, ML-based VIS+IR, and ML-based IR-single, have been developed





and utilized to retrieve CBHs from GEO H8/AHI data. The two physics-based
algorithms utilize cloud top and optical property products from AHI as input
parameters to retrieve high spatial-temporal resolution CBHs, with operations limited
to daytime. In contrast, the ML-based VIS+IR model and IR-single model algorithms
use the matched joint CloudSat/CALIOP CBH product as true values for building RF
prediction models. Notably, the ML-based IR-single algorithm, which relies solely on
infrared band measurements, can retrieve CBH throughout the day.
The accuracy of CBHs retrieved from the four independent algorithms is verified
using the joint CloudSat/CALIOP CBH products for the year 2017. The GEO IDPS
algorithm shows an R of 0.62 and an RMSE of 2.642 km. The GEO CLAVR-x
algorithm provides more accurate CBHs with an R of 0.647 and RMSE of 2.91 km.
After filtering samples with optical thickness less than 1.6 and brightness temperature
(at 11 μm band) greater than 281 K, the ML-based VIS+IR and ML-based IR-single
algorithms achieve higher accuracy with an R(RMSE) of 0.922(1.214 km) and
0.911(1.415 km), respectively. This indicates strong agreement between the two
ML-based CBH algorithms and the CloudSat/CALIOP CBH product.
However, in stark contrast, the results from the physics-based algorithms are
superior to those from the ML-based algorithms (with R and RMSE of 0.592/2.86 km
and 0.385/3.88 km, respectively) when compared with ground-based CBH
observations such as lidar and cloud radar. In the comparison with the cloud radar at
Beijing Nanjiao station in 2017, the R of the GEO CLAVR-x algorithm is 0.573,
while the R of the GEO IDPS algorithm is 0.515. Meanwhile, notable differences are
observed in the CBHs from both ML-based algorithms. Similar conclusions are also
evident in the 2-day comparisons at Yunnan Lijiang station.
The CBH results from the two ML-based algorithms (R > 0.91) can likely be
attributed to the use of the same training and validation dataset source as the joint
CloudSat/CALIOP product. However, this dataset has limited spatial coverage and
small temporal variation, potentially limiting the representativeness of the training
data. In contrast, the GEO CLAVR-x algorithm demonstrates the best performance
and highest accuracy in retrieving CBH from geostationary satellite data. Notably, its
results align well with those from ground-based lidar and cloud radar during the
daytime. However, both physics-based methods, utilizing CloudSat CPR data for
regression, struggle to accurately retrieve CBHs below 1 km, as the lowest 1 km
above ground level of this data is affected by ground clutter.





Additionally, despite utilizing the same physics principles in spaceborne and
ground-based lidar/radar CBH algorithms, the previous study (Thorsen et al., 2011)
has highlighted differences in profiles between them. Therefore, this factor could
contribute to the relatively poorer results in CBH retrieval by ML-based algorithms
compared to ground-based lidar and radar. The analysis and discussion above suggest
that ML-based algorithms are constrained by the size and representativeness of their
datasets. Therefore, in scenarios involving a large time scope, such as climate
research, it is more reasonable to opt for physics-based cloud base height algorithms.
Ideally, if more spaceborne cloud profiling radars with different passing times
(covering all day) can be included in the training dataset, the promising ML technique
will certainly generate a higher quality CBH product with more comprehensive
observations. The CBH product using ML-based algorithms should continue to be
improved in future work. At present, we will focus on developing physics-based
algorithms for cloud base height for the next generation of geostationary
meteorological satellites, to support the application of these products in weather and
climate domains.
Besides, at night, current GEO satellite imaging instruments encounter
challenges in accurately determining CBH due to limited or absent solar illumination.
Because it is unable to retrieve cloud optical depth in the visible band, the current
method faces limitations. However, there is potential for enhanced accuracy in
deriving cloud optical and microphysical properties, as well as CBH, by incorporating
the Day/Night Band (DNB) observations during nighttime in the future (Walther et al.,
2013).



*Data availability.* The authors would like to acknowledge NASA, JMA, University of
Colorado, and NOAA for freely providing the MODIS
(https://ladsweb.modaps.eosdis.nasa.gov/search), CloudSat/CALIOP
(https://www.cloudsat.cira.colostate.edu/), Himawari-8 (ftp.ptree.jaxa.jp), and GFS
NWP (ftp://nomads.ncdc.noaa.gov/GFS/Grid4) data online, respectively.


*Author contributions.* MM proposed the essential research idea. MW, MM, JL, HL,



BC, and YL performed the analysis and drafted the manuscript. ZY and NX provided
useful comments. All the authors contributed to the interpretation and discussion of
results and the revision of the manuscript.


*Competing interests.* The authors declare that they have no conflict of interest.


*Acknowledgements.* The authors would like to acknowledge NASA, JMA, University
of Colorado, and NOAA for freely providing satellite data online, respectively. The
authors thank NOAA, NASA, and their VIIRS algorithm working groups (AWG) for
freely providing the VIIRS cloud base height algorithm theoretical basic
documentations (ATBD). In addition, the authors appreciate the power computer tools
developed by the Python and scikit-learn groups (http://scikit-learn.org). Besides the
authors also thank Rundong Zhou and Pan Xia for drawing some pictures of this
manuscript. Last but not the least, the authors sincerely thank Prof. Yong Zhang and
Prof. Jianping Guo for freely providing cloud base height results retrieved by
ground-based cloud radar at Beijing Nanjiao station. This work was supported partly
by the Natural Science Foundation of Shanghai (No. 21ZR1419800), the Guangdong
Major Project of Basic and Applied Basic Research (Grant 2020B0301030004),
National Natural Science Foundation of China under Grants 42175086 and U2142201,
FengYun Meteorological Satellite Innovation Foundation under Grant
FY-APP-ZX-2022.0207, Innovation Group Project of Southern Marine Science and
Engineering Guangdong Laboratory (Zhuhai) (No. 311022006), and the Science and
Technology Planning Project of Guangdong Province (2023B1212060019). We
would like to thank the editor and anonymous reviewers for their thoughtful
suggestions and comments.




**Appendix A**

Based on the previously discussed description of two physics-based cloud base height (CBH) retrieval algorithms (GEO IDPS and GEO CLAVR-x retrieval algorithms), cloud products such as cloud top height (CTH), effective particle radius ($R_{\text{eff}}$), and cloud optical thickness ($D_{\text{COT}}$) will be utilized in both algorithms. To validate the reliability of these cloud products derived from the Advanced Himawari Imager (AHI) aboard the Himawari-8 (H8), a pixel-by-pixel comparison is conducted with analogous MODIS Collection-6.1 Level-2 cloud products. Both Aqua and Terra MODIS Level-2 cloud products (MOD06 and MYD06) are accessible for free download from the MODIS official website. For verification purposes, the corresponding Level-2 cloud products from January, April, July, and October of 2018 are chosen to assess CTH, $D_{\text{COT}}$, and $R_{\text{eff}}$ retrieved by H8/AHI.

Fig. S2 (in the supplementary document) shows the spatially-temporally matched case comparisons of CTH, $D_{\text{COT}}$ and $R_{\text{eff}}$ from H8/AHI and Terra/MODIS (MYD06) at 03:30 UTC on January 15, 2018. It can be seen that the CTH, $D_{\text{COT}}$ and $R_{\text{eff}}$ from H8/AHI are in good agreement with the matched MODIS cloud products. However, there are still some differences in $R_{\text{eff}}$ at the regions near 35°N, 110°E in Figs. S2d and S2c. The underestimated $R_{\text{eff}}$ values from H8/AHI relative to MODIS have been reported in previous studies. (Letu et al., 2019) compared the ice cloud products retrieved from AHI and MODIS, and concluded that the $R_{\text{eff}}$ from both products differ remarkably in the ice cloud region and the $D_{\text{COT}}$ from them are roughly similar. However, the $D_{\text{COT}}$ from AHI data is higher in some areas. Looking again at the cloud optical thickness that at the same time, the slight underestimation of H8/AHI $D_{\text{COT}}$ can be found in Figs. S2e and S2f. Fig. S3 (in the supplementary document) shows another case at 02:10 UTC on January 15, 2018. Despite of the good consistence between H8/AHI and MODIS cloud products, there are slight differences in CTH in the area around 40°S–40.5°S, 100°E–110°E in Figs. S3a and S3b. Besides, as shown in Fig. S2, there are still underestimations in the $R_{\text{eff}}$ of H8/AHI.

To further compare and validate these three H8/AHI cloud products, the spatially-temporally matched samples from H8/AHI and Aqua/Terra MODIS in four months of 2018 are counted within the three intervals of 0.1 km (CTH), 1.0 μm ($R_{\text{eff}}$), and 1 ($D_{\text{COT}}$) in Fig. S4 (in the supplementary document). The corresponding mean absolute error, mean bias error, root mean square error and correlation coefficient ($R$) values are also calculated and marked in each subfigure. As can be seen, the $R$ of





CTH is around 0.75 in all four months and is close to 0.8 in August. The results of
$D_{COT}$ show the highest $R$, reaching above 0.8. In contrast, the underestimation trend in
$R_{eff}$ is also shown in this figure. These different consistencies between two
satellite-retrieved cloud products may be attributed to: (1) different spatial-temporal
resolutions between H8/AHI and MODIS; (2) different wavelength bands, bulk
scattering model, and specific algorithm used for retrieving cloud products; (3)
different view zenith angle between GEO and low-earth-orbit satellite platforms (Letu
et al., 2019). In addition, other external factors such as surface type also can affect the
retrieval of cloud product. However, according to Fig. S4, the bulk of the analyzed
samples are still around the 1:1 line, indicating the good quality of H8/AHI cloud
products.

**Appendix B**
The ML-based visible (VIS)+infrared (IR) model algorithm mentioned above
uses 230 typical variables (see Table 1) as model predictors, and the importance
scores of top-30 predictors are ranked in Fig. S5 (in the supplementary document). It
can be seen that the most important variables are CTH and cloud top temperature, and
$D_{COT}$ is an important or sensitive factor affecting these two quantities. A sensitivity
test is also performed to further investigate the potential influence of $D_{COT}$ on the
CBH retrieval by the VIS+IR model (see Table S1 in the supplementary document).
From Fig. S7a, we find that the samples with $D_{COT}$ lower than 5 cause the relatively
large CBH errors compared with the matched CBHs from the joint CALIPSO
(Cloud-Aerosol Lidar and Infrared Pathfinder Satellite Observation)/CloudSat
product.
According to the results in this Fig. S7b, we may filter the samples with
relatively small $D_{COT}$ to further improve the accuracy of CBH retrieval by the VIS+IR
model (see Table S1). Fig. B3b shows that after filtering the samples with the $D_{COT}$
less than 1.6, the $R$ increases from 0.895 to 0.922, implying a better performance of
CBH retrieval. According to the ranking of predictor importance (see Fig. S6 in the
supplementary document), we also conduct another sensitivity test on the BT
observed by H8/AHI IR Channel-14 (Cha14) at 11 μm, which plays an important role
in the IR-single model. Fig. S7c shows that the BT values of H8/AHI Channel-14
ranges from 160 K to 316 K, and the samples with BT higher than 300 K show large
CBH errors. Similarly, by filtering the samples with BT higher than 281 K, we can get



a better IR-single model algorithm for retrieving high-quality CBH (see Table S2 in
the supplementary document). Fig. S7d also proves that the $R$ value increases from
0.868 to 0.911.

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





963                                    **Tables and Figures**


**Table 1.** Predictand and predictor variables for both visible (VIS)+infrared (IR) model
and IR-single regression model training, which are divided according to the different
predictor variables from satellite and NWP data

| Predictand | Cloud base height from the joint CloudSat/CALIPSO product | |
|---|---|---|
| **Predictor [satellite measurements]** | IR-single | BT(3.9μm), BT(6.2μm), BT(6.9μm), BT(7.3μm), BT(8.6μm), BT(9.6μm), BT(10.4μm), BT(11.2μm), BT (12.4μm), BT (13.3μm), BTD(11.2–12.4μm), BTD(11.2–13.3μm) [Unit = K], Air Mass (1/cos(VZA)), View azimuth angles [Unit = degree], Cloud top height from H8/AHI [unit: m], Cloud top temperature from H8/AHI [unit: K] |
| | VIS+IR | Ref(0.47μm), Ref(0.51μm), Ref(0.64μm), Ref(0.86μm), Ref(1.64μm), Ref(2.25μm), BT(3.9μm), BT(6.2μm), BT(6.9μm), BT(7.3μm), BT(8.6μm), BT(9.6μm), BT(10.4μm), BT(11.2μm), BT(12.4μm), BT(13.3μm), BTD(11.2–12.4μm), BTD(11.2–13.3μm) [Unit = K], Air Mass(1/cos(VZA)), Air Mass(1/cos(SZA)), View/Solar Azimuth angles [Unit = degree], Cloud top height from H8/AHI [unit: m], Cloud top temperature from H8/AHI [unit: K] |
| **Predictor [GFS NWP]** | IR-single/ VIS+IR | Altitude profile (from surface to about 21 km, 67 layers) [unit: m], Temperature profile (from surface to about 21 km, 67 layers) [unit: K], Relative humidity profile (from surface to about 21 km, 67 layers) [unit: %], Total precipitable water, Surface temperature [unit: K] |
| **Predictor [other]** | IR-single/ VIS+IR | Surface elevation [unit: m] |

Notes: VZA = view zenith angle [unit: degree]; SZA = solar zenith angle [unit:
degree]











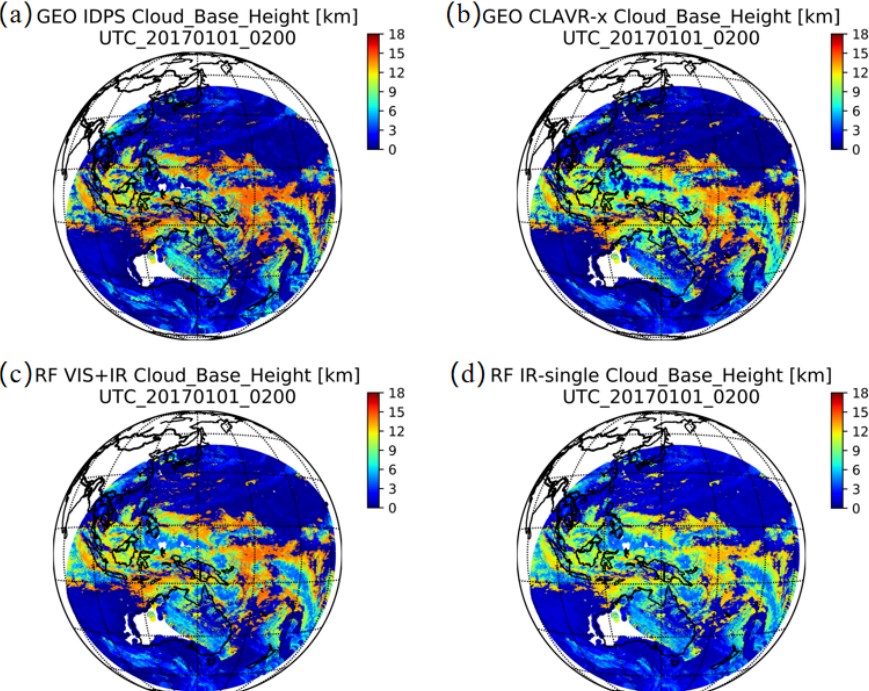


**Figure 1.** Comparison of full disk CBH results retrieved by the four-independent algorithms at 02:00 UTC on January 1, 2017. (a) GEO IDPS algorithm, (b) GEO Clouds from AVHRR Extended (CLAVR-x) algorithm, (c) ML-based (RF, random forest) VIS+IR algorithm and (d) ML-based (RF) IR-single algorithm.







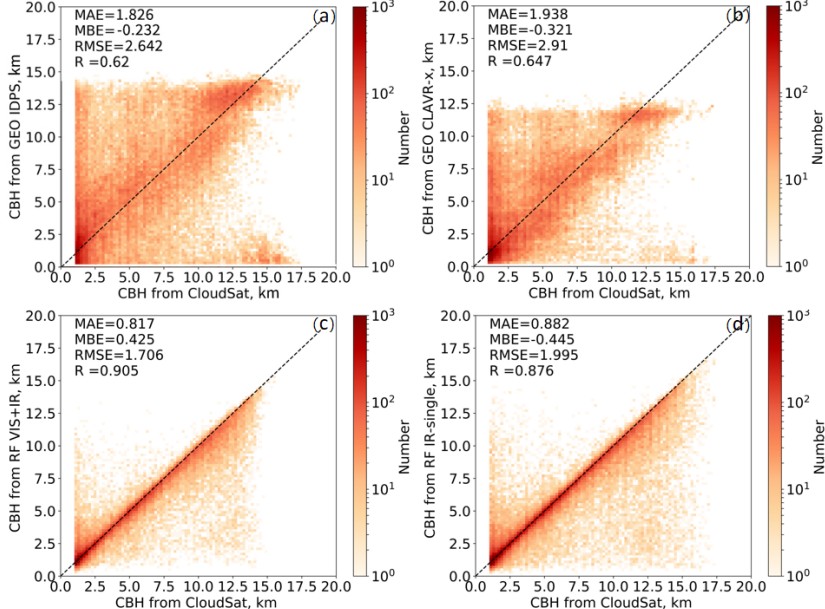

**Figure 2.** Density distributions of CBHs retrieved from (a) GEO IDPS, (b) GEO
CLAVR-x, (c) VIS+IR and (d) IR-single algorithms compared with the CBHs from
the joint CloudSat/CALIPSO product (taken as true values) in 2017. The mean
absolute error (MAE), mean bias error (MBE), root mean square error (RMSE) and R
are listed in each subfigure where the difference exceeds the 95% significance level (p
< 0.05) according to the Pearson's $\chi2$ test.


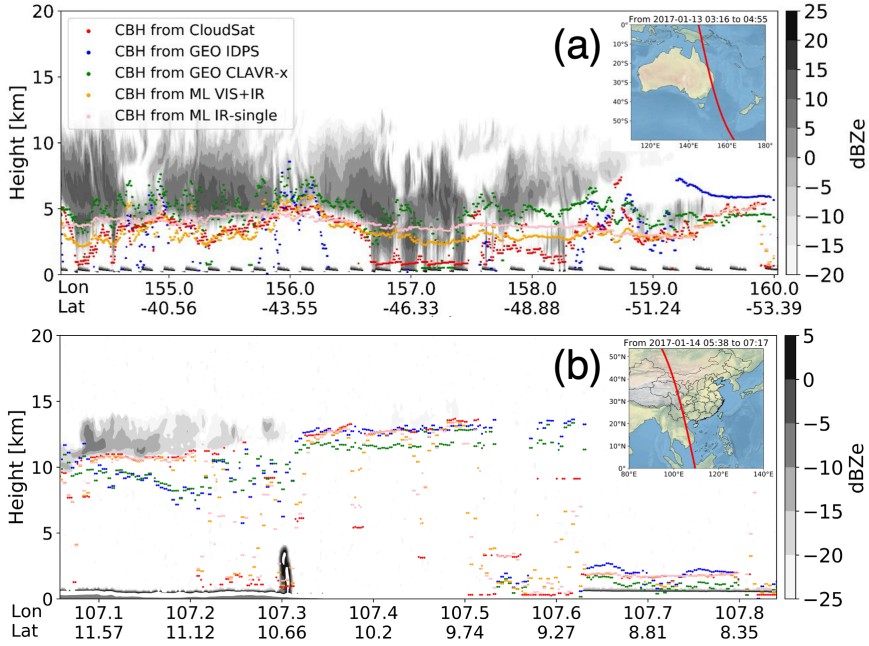


**Figure 3.** Inter-comparisons of CBH products retrieved by CloudSat (red solid circle),
the GEO IDPS algorithm (blue solid circle), the GEO CLAVR-x (green solid circle),
the ML-based VIS+IR model algorithm (orange solid circle), and the ML-based
IR-single model algorithm (pink solid circle) at (a) 03:16–04:55 UTC on January 13,
2017 (a) and (b) 05:38–07:17 UTC on January 14, 2017. The black and gray colormap
represents the matched CloudSat radar reflectivity.















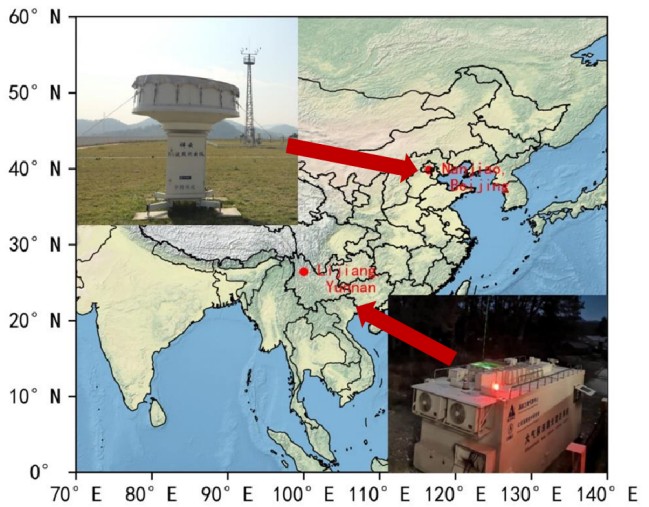


**Figure 4.** Geographical locations and photos of lidar and cloud radar at Yunnan
Lijiang and Beijing Nanjiao stations.






















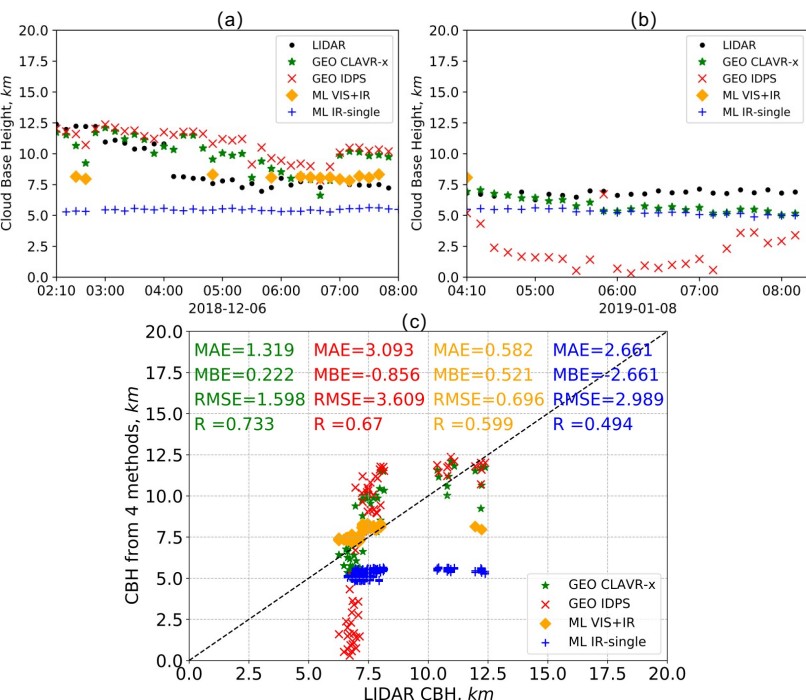


**Figure 5.** Comparisons of the CBHs from the ground-based lidar measurements
(black solid circle) at Yunnan Lijiang station and the four GEO satellite retrieval
algorithms, namely the GEO IDPS (red cross symbol), the GEO CLAVR-x (green
solid asterisk), the ML-based VIS+IR model (orange solid diamond) and the
ML-based IR-single model (blue plus sign) algorithms. Fig 6a and 6b show the time
series of CBHs from lidar and the four GEO satellite retrieval algorithms on
December 6, 2018 and January 8, 2019, respectively. Fig 6c shows the scatterplots of
CBH samples from the lidar measurements and the four retrieval algorithms.













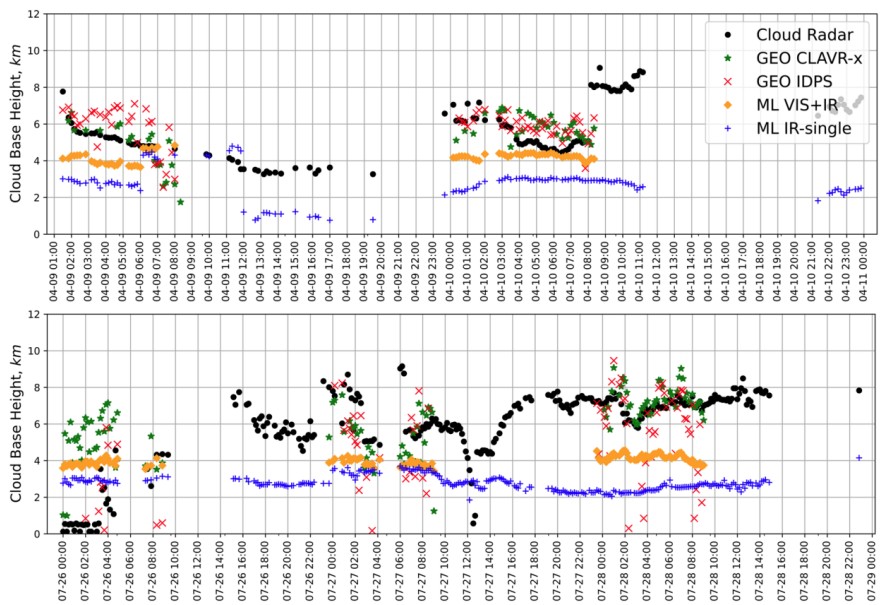


**Figure 6.** Same as Fig. 5, but for the CBH sample results from the cloud radar at
Beijing Nanjiao station (black solid circle) on April 9–10, 2017 (top panel) and July
26–28, 2017 (bottom panel).

















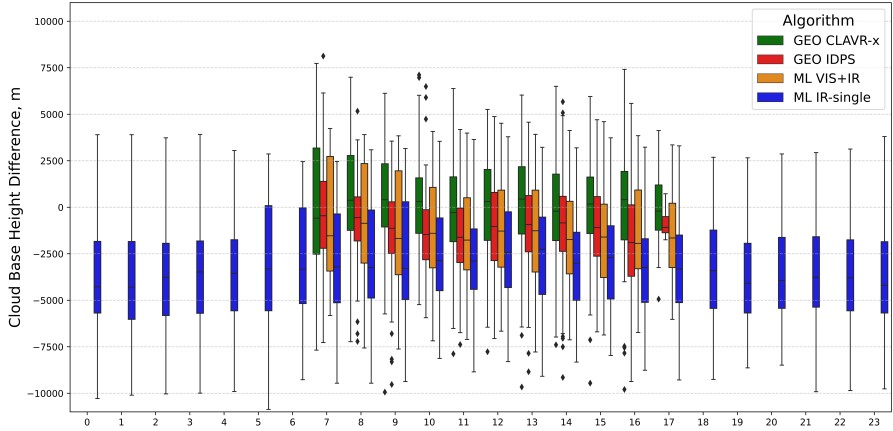


**Figure 7.** Box plots of the hourly CBH errors of four GEO satellite retrieval
algorithms (GEO IDPS, GEO CLAVR-x, ML-based VIS+IR and ML-based IR-single)
relative to the CBHs from the cloud radar at Beijing Nanjiao station in 2017. The box
symbols signify the 25th, 50th and 75th percentiles of errors. The most extreme
sample points between the 75th and outlier, and the 25th percentiles and outliers are
marked as whiskers and diamonds, respectively. Except for the period between 7 and
17 UTC (local time), the three algorithms of GEO CLAVR-x, GEO IDPS, and ML
VIS+IR are unavailable due to the lack of reflected solar radiance measurements.




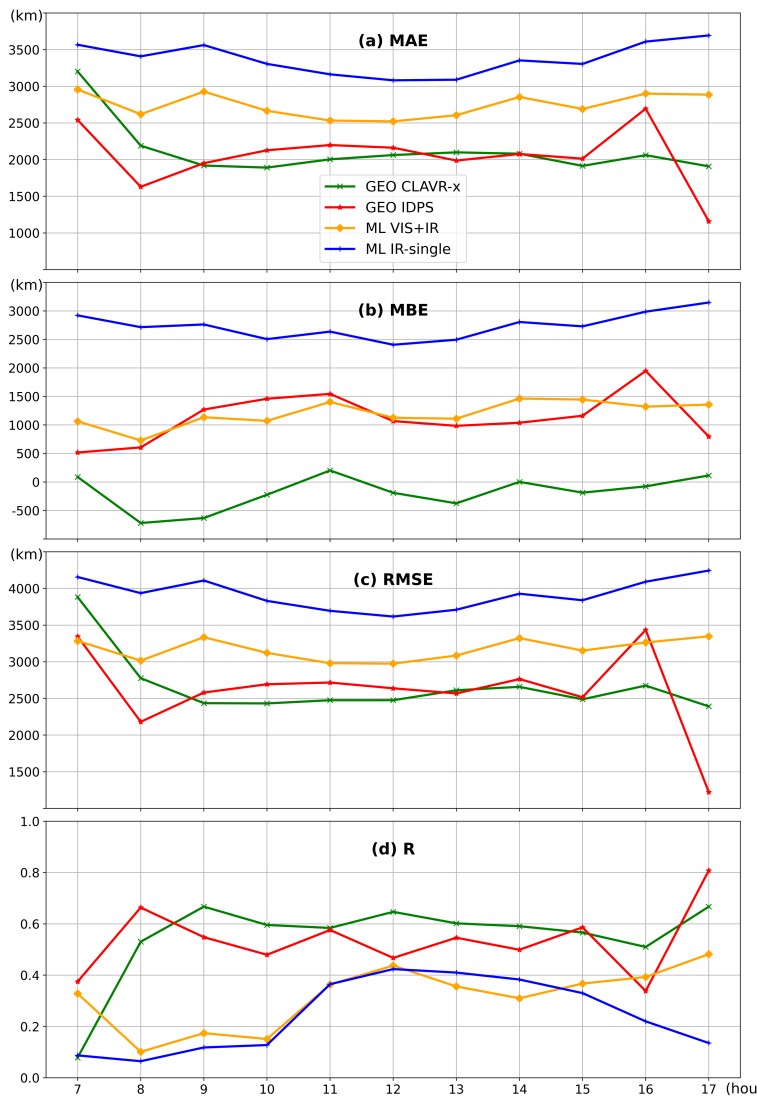


**Figure 8.** Comparisons of hourly (a) MAE, (b) MBE, (c) RMSE, and (d) R of CBH

(relative to the CBHs from the cloud radar at Beijing Nanjiao station) from 07 to 17

(local time) between four retrieval algorithms (GEO IDPS, GEO CLAVR-x,

ML-based VIS+IR and ML-based IR-single) in 2017.

1131

1132

1133



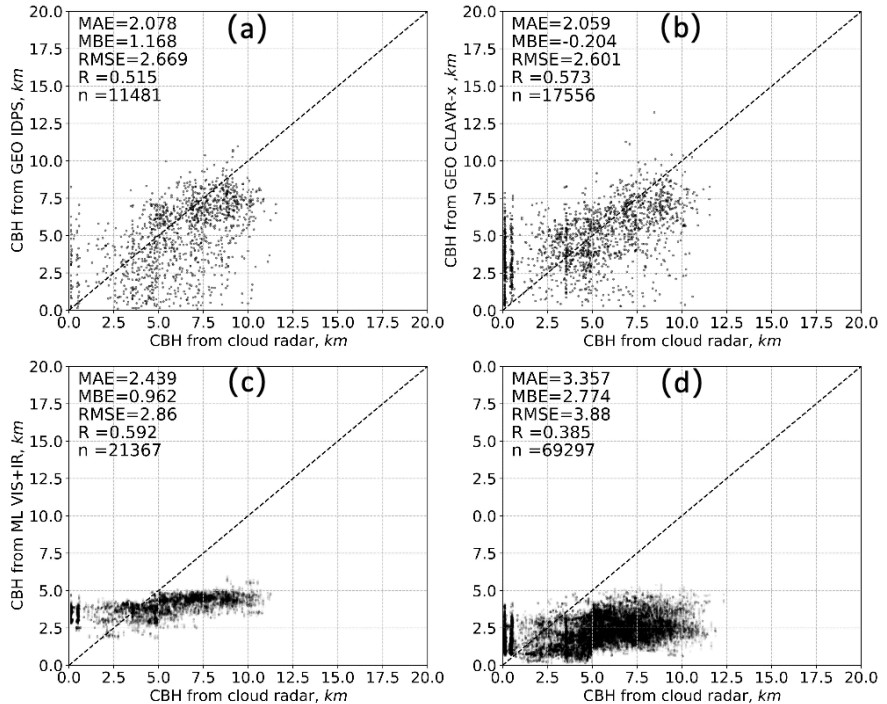

**Figure 9.** Comparisons between the CBHs from the cloud radar at Beijing Nanjiao

station and the matched CBHs from the four retrieval algorithms (GEO IDPS, GEO

CLAVR-x, ML-based VIS+IR and ML-based IR-single) in 2017.