# Peer review of "Technical note: Applicability of physics-based and machine-learning-based"

_EGUsphere, 2024_

## Author Response (AR1)

**Reviewer 1**

Dear editor, dear authors,

This manuscript compares two types of models for the retrieval of the cloud base height (CBH) and investigate in more depth the retrieval of its diurnal cycle. The authors use geostationnary satellite data from the Himawari-8 Advanced Himawari Imager (AHI) instrument to describe two physics-based (IDPS CBH and CLAVR-x CBH algorithms) and two machine learning (RF VIS+IR and RF IR-single algorithms) CBH retrieval methods. These methods are first introduced, then evaluated against Calipso/Cloudsat joint CBH retrievals and eventually compared to surface-based lidar and radar measurements from two instruments in China. The physics-based algorithms are built on cloud properties such as cloud optical thickness, cloud phase or cloud top height from the corresponding cloud products of the geostationnary satellite. On the other hand, the machine learning algorithms are built on measurements from infrared and visible bands, along with NWP data including for example temperature profiles or precipitable water. The paper investigates interesting questions regarding the benefits and drawbacks of the two types of models presented. However, the ambiguity about the Calipso/Cloudsat data source on which the machine learning models are trained and then evaluated alongside the two physics-based models renders the evaluation ambiguous and could benefit from some clarifications. The comparison to surface-based measurements allows for further comparison and evaluation of the four algorithms. Furthermore, additional details about the diurnal cycle of CBH in the observations and in the retrievals should be emphasised clearly in the corresponding sections. Only the ML-based IR-single method is able to retrieve CBH during nighttime, hindering the evaluation of the full diurnal cycle of CBH from the four retrieval methods.

Detailed comments are included in the following sections, followed by more technical comments.

**Specific comments:**

   • Training and evaluation Cloudsat/Calipso datasets:

◦ In section 3.3, what is the reason for the different amounts of points in the dataset between the two methods? Just a clarification that it stems from the fact that only daytime data can be used for the VIS+IR method should be added here (if that is actually the reason).

Answer: Thank you for your suggestion. The reason for the different amount of points between the two methods is that the VIS band only can observe reflectance during the daytime. We have added a sentence "*Note that the reduced data amount is because only daytime data can be used for the VIS+IR method training.*" in Lines 357-358 to further explain the different amounts of points.

◦ The random splitting of the data for training and testing might lead to spurious correlation when evaluating the method due to auto-correlation between samples. Furthermore, are the two datasets split in a way that the testing datasets contain the same samples? Splitting the data according to spatiotemporality could circumvent such issue, but as the dataset is built on a single year it might prove difficult to properly investigate here.

Answer: Thank you for your suggestion. This is an issue we are well aware of, and similar methods have been used in previous studies (see Figure 3 from Min et al., 2020). We constructed the dataset using data from 2017 and applied the `split_train` function in Python to allocate 70% for training and 30% for validation, ensuring that the sample is independent. We have added a sentence to further explain this issue "*As in earlier study (Min et al. 2020), we also used 70% of the matched data for training and 30% of an independent sample for validation.*" in Lines 390-392.

Reference:

Min Min, Jun Li, Fu Wang, Zijing Liu, W. Paul Menzel, 2020. Retrieval of cloud top properties from advanced geostationary satellite imager measurements based on machine learning algorithms [J]. Remote Sensing of Environment, 239: 111616, doi: 10.1016/j.rse.2019.111616

◦ I did not quite understand in lines 351-353 if you mean that the training datasets are also used during validation with the Cloudsat/Calipso product?

◦ Generally, I find it unclear on which samples the evaluation is built compared to the training samples for the ML-based algorithms. Details should be

included either in section 3.3 or at the beginning of section 4 because it is crucial in the comparison of the different algorithms.

Answer: Thank you for your suggestion. This is something we didn't clearly explain. We believe this issue is similar to the previous one, so please refer to our earlier response.

∘ The scores of the two ML-based methods on their respective (or maybe on a common) testing dataset should be included if different from the ones presented in Figure 2.

Answer: Thank you for your suggestion. The testing or validation dataset for two ML-based methods is the same. More explanation can be found in the answers to the previous questions.

• Results section:

∘ In section 4.1, it might be interesting to include subsections to group the different aspects evaluated during the comparison to the Cloudsat/Calipso CBH retrievals to improve readability. I would suggest subsections as follow (titles might obviously need some rewording): 4.1.1 Joint scatter plots, 4.1.2 Test case. However, the last paragraph (lines 446-457) is misplaced and should be included in section 3.3 with the description of the ML-based algorithms. The first paragraph of the section (L363-372) could even be placed as an introduction to section 4.

Answer: Thank you for your suggestion. The last paragraph is replaced in the last paragraph of section 3.3. In addition, we have added subsections in section 4.1 according to your advice.

∘ Similar comment can be made regarding the structure of section 4.2. Seperating the evaluation experiments would improve readability. For example, one could put the use cases showcased in figures 5 and 6 in a first section and then the results for the whole year of 2017 (figures 7-9).

Answer: Thank you for your suggestion. The first subsection that put the use cases showcased in figures 5 and 6 is named 4.2.1 Comparison of CBH retrievals from ground and satellite data, while the second subsection that the results for the whole

year of 2017 (figures 7-9) is named 4.2.2 Diurnal cycle analysis of CBH retrieval accuracy.

  ◦ In the subsequent analysis of the CBH retrievals, the diurnal cycle characteristics of the CBH are mentioned but never clearly explained or put in context with respect to the measurements or time series presented. Additionally, only the ML-based IR-single method is actually able to retrieve CBH during nighttime, making the evaluation of the full diurnal cycle of the retrieved CBH from the different methods impossible. This should be highlighted in the manuscript somewhere.

Answer: Thank you for your suggestion. Agree, we think the diurnal cycle characteristics of the CBH are important. We have added a paragraph to highlight this issue "*As well known, the diurnal variation of cloud base height is primarily influenced by solar heating, causing the cloud base to rise in the morning and reach its peak by midday. As the surface cools in the afternoon and evening, the cloud base lowers, playing a crucial role in weather patterns and forecasting (Zheng et al. 2020). Therefore, it is essential to rigorously compare the ML-based algorithm with ground-based observations to determine its ability to adapt to the daily variations in cloud base height caused by natural factors.*" in Line 531.

Reference

Zheng, Y., Sakradzija, M., Lee, S.-S., and Li, Z.: Theoretical Understanding of the Linear Relationship between Convective Updrafts and Cloud-Base Height for Shallow Cumulus Clouds. Part II: Continental Conditions, J Atmos Sci, 77, 1313-1328, 10.1175/jas-d-19-0301.1, 2020.

  ◦ In section 4.2, is there a particular reason for the choice of the dates of December 6, 2018, and January 8, 2019 to perform the validation? Similar comment is valid for the second use case with the Beijing Nanjiao station. A description of the cloud scene or of the characteristics of the measured CBH time series would be a great addition to give further background ahead of the comparison.

Answer: Thank you for your suggestion. Due to the small sample size, we only have 2 days of cloud data for comparison. We have added a sentence to explain this issue "*In fact, this lidar was primarily used for the calibration of ground-based lunar radiation*

*instruments. During the two-month observation period (from December of 2018 to January of 2019), it was always operated only under clear sky conditions, resulting in the capture of cloud data on just two days.*" in Line 494.

The following figure shows the experimental environment and test instruments in Lijiang Station.

[Figure]

• Conclusions and discussion:

◦ Paragraph L611: As mentioned previously, the potential auto-correlation between samples in the training and testing datasets can inflate the performance skill. A comment about how the data could be split to evaluate the generalisation skill to unseen locations or time periods could be included here. Following the subsequent analysis of the performance of the different ML-based algorithms, either the main limitation is the representativeness of the training data, or the potential overfitting of the models to the modality of the training dataset.

Answer: Thank you for your suggestion. The question of training and validating the data set has been explained in line 391, and we used independent training and validation datasets. We think that our model developed in this study is not overfitted, and we believe the main reason is that the training samples from Cloudsat/Calipso do not adequately represent the daily cycle characteristics of the cloud base. This point have been proved by using comparisons with the ground-based cloud observation data, which is the key finding of this study.

In fact, other scholars (private communication) working on ML-based cloud base inversion have encountered similar issues, but they have not performed long-term data testing and analysis. This is the core of our manuscript, which highlights the

limitations of using Cloudsat/Calipso as a ground truth for training cloud base or other attribute models.

  ◦ Paragraph L629-636: A short comment on the avenue of research with Physics-informed ML could be interesting. ML models building on known or trusted physical relationships could potentially bridge the gap in the case of problematic and challenging retrievals for cloud properties for example.

Answer: Thank you for your suggestion. Agree, we have added a new paragraph "*Particularly, exploring the joint ML-physics-based method presents a promising direction, which can address the complexities and challenges in retrieving cloud properties. By integrating established physical relationships into ML models, we can potentially enhance the accuracy and reliability of predictions. This approach not only leverages the strengths of both physics-based models and data-driven techniques but also offers a pathway to more robust and interpretable solutions in atmospheric sciences.*" here to comment this issue under your suggestion.

  • L40-42: Mention on which dataset are the ML methods outperforming the other two and for which method is the R score is given.

Answer: Thank you for your suggestion. This R score represents the optimal method using VIS+IR training data from Figure 2. We have added "the optimal method is …" here to explain this issue.

  • L45 and L53: Same as previous comment, to which method is the R score referring?

Answer: Thank you for your suggestion. It is GEO CLAVR-x method. We think it is meaningless to give this name (CLAVR-x) in abstract without background, so I can only use the optimal method instead. We have highlighted "the optimal physics-based algorithm" in Line 41.

  • The comparison to MODIS data included in Appendix A and the supplementary materials is interesting to gain insights on the joint distributions of the cloud properties and evaluate the Himawari-8 AHI instrument. However, if I am not mistaken, the methods presented in Baker at al. (2011) are based on data from VIIRS

instrument. A comparison to the corresponding cloud product could also be of interest to assess the portability/transferability of the algorithms.

Answer: Thank you for your suggestion. We also explored and attempted past comparisons but found that the correlation was not strong. While our algorithm shows a slightly better correlation than that of VIIRS, the RMSE is higher (Compared with Cloudsat/Calipso). The official VIIRS algorithm is not available to us, and it only approximates the CLARV-x algorithm, which cannot be exactly replicated. Additionally, it depends on several upstream cloud retrieval products, such as cloud detection and cloud top height. As a result, evaluating the final consistency is challenging, and it remains unclear which product should be considered the closest to the true value.

• L338-339: Several hyperparameters of the random forest models are listed but only the chosen number of estimators and the maximum depth are mentioned.

Answer: Thank you for your suggestion. The random forest model can only debug these few hyperparameters, and I have tested and listed them all in this section.

• L447: A reference or a sentence about how the importance scores are computed would be useful.

Answer: Thank you for your suggestion. We have added a sentence and a reference "*In a Random Forest model, feature importance indicates how much each input variable contributes to the model's predictive accuracy by measuring the decrease in impurity or error when the feature is used to split data (Gregorutti et al. 2017).*" in Line 370.

**Additional specific/technical comments:**

General comment: The spatial and temporal adjectives when combined should write as spatiotemporal (eg. L580) or spatiotemporally (eg. L120, L582).

Answer: Thank you for your suggestion. We have corrected the corresponding words in the entire text.

Across the manuscript, when an acronym is introduced first, you should refer to it in the rest of the manuscript. For instance, cloud optical thickness is first introduced at line 128-129 without the acronym, then at line 194-195 with the acronym and again introduced at line 268 with two different acronyms. Another example is the cloud mask (CLM) mentioned line 194 and at line 198 in plain text. Similar comments follow for example for cloud geometric thickness (line 264) and cloud top height (line 265). Overall, these inconsistencies render reading sometimes a bit difficult with the constant switching between plain text and acronyms. The authors should make sure that throughout the manuscript, the acronyms are introduced first and then used consistently. The acronyms for the cloud properties could be introduced at line 71 for example.

Answer: According to your suggestion, we checked again and made the correct changes to the acronyms CBH, COT, CTH, CLM and CGT throughout the entire text. For example, we first define the cloud top height with the acronym CTH, the two acronyms COT / $D_{COT}$ of cloud optical thickness in line 71, and use these acronyms in the latter text.

Generally, when used in the text and not in parentheses, the figure mention should be included in full: "Figure 1 displays …" (L365), "Figure 2 presents …" (L373), "Similar to Figure 5, Figure 6 …" (L529)

Answer: Following your nice suggestion, we have revised the figure mention in these sentences in the entire text.

L84: "… ramifications of clouds …".

Answer: Thanks for your nice advice. We have replaced the word 'cloud' with the word 'clouds'.

L114: The reference needs to be properly included: "A recent study by Yang et al. (2021) …".

Answer: Thanks for your nice advice. We have revised the sentence in L128.

L119: The reference needs to be properly included: "For instance, Wang et al. (2012) …".

Answer: Thanks for your nice advice. We have revised the sentence in L133.

L124: "… corresponding CBH …".

Answer: Thanks for your nice advice. We have changed to the word 'corresponding'.

L126: The reference is not included in full for Hutchinson et al. (2003? 2006? or both?).

Answer: Thanks for your suggestion. We have already added the years 2002 and 2006 for Hutchinson et al. (2002 and 2006).

Hutchison, K. D.: The retrieval of cloud base heights from MODIS and three-dimensional cloud fields from NASA's EOS Aqua mission, Int J Remote Sens, 23, 5249-5265, 10.1080/01431160110117391, 2002.

Hutchison, K., Wong, E., and Ou, S. C.: Cloud base heights retrieved during night‐time conditions with MODIS data, Int J Remote Sens, 27, 2847-2862, 10.1080/01431160500296800, 2006.

L134: Drop the "previous".

Answer: Following your advice, we have dropped the word 'previous'.

L143: Please include the reference for the ERA5 dataset.

Answer: We have added the website address (https://cds.climate.copernicus.eu/cdsapp#!/search?type=dataset) as the reference for the ERA5 dataset.

L139-149: The references for Tan et al. (2020), Lin et al. (2022) and Tana et al. (2023) are included twice in the respective sentences describing their methods. The references at the end of these sentences can be omitted.

Answer: Thank you for your suggestion. The references for Tan et al. (2020), Lin et al. (2022) and Tana et al. (2023) at the end of these sentences have been omitted.

L189: "… facilitate …".

Answer: We have revised it.

L210: "… global high-quality …".

Answer: We have revised it.

L215: Include the references for the MODIS Cloud product Collection 6.1: Platnick et al. (2017).

Answer: We have revised it.

L274: "… multi-layered cloud systems …".

Answer: We have revised it.

L289: Do you mean the GOES-R geostationary satellite imager?

Answer: Yes, you are right. And we have revised it.

L320: "… regression Random Forest model …".

Answer: We have revised it.

The equations 2-6 could be included in a different section, for example at the beginning of the method or result sections, as the metrics are used for comparing all the methods and do not only pertain to the ML-based algorithms.

Answer: Thanks for your suggestion. Yes, you are right. We have added a new section of 3.4 Evaluation method for all the algorithms.

L325-326: Include the formulas for the air mass predictors as a mathematical equation object.

Answer: Thanks for your suggestion. We assumed the readers would comprehend these two variables in this format.

L379: "Seaman et al. (2017) …".

Answer: Thanks for your suggestion. We have revised it.

L385: Valid for other instances but the metrics should be reported with the same precision consistantly.

Answer: Thanks for your suggestion. This may confuse you. We have added a sentence "........*CBH algorithms, compared with VIIRS CBH product (Seaman et al. 2017).*" to further explain the comparison with VIIRS CBH product.

L447: "… importance scores …".
Answer: We have revised it.

L521-523: Is the verb (ie. an "are") missing in this sentence?
Answer: Yes, you are right. We have been revised it.

L532-534: Monthly aggregated results are stated to be in the supplements but are not included. Furthermore, this sentence should be included earlier when the yearly dataset is presented, namely line 508.
Answer: Thanks for your suggestion. We have moved and changed this sentence to the Line 528 "*Due to the density of points in the one-year time series, the point-to-point CBH comparison results for the entire year are not displayed here (monthly results are shown in the supplementary document), we only show 4 days results in the following Figure 6.*"

L580: Data from 2019 is also used (comparison with Lijiang station).
Answer: Thanks for your suggestion. Yes, you are right. We have revised this to "*from 2017 to 2019*" in Line 607.

L593: Clarify that the ML-based IR-single algorithm can retrieve cloud base during both day and night, "throughout the day" is a bit ambiguous.
Answer: Thanks for your suggestion. We have revised it.

L603-605: Clarifiy for which methods among the physics-based and the ML-based are the R and RMSE metrics obtained.
Answer: We have revised it. We have changed the sentence to '...the results from the physics-based algorithms (with R and RMSE of 0.592/2.86 km) are superior to those from the ML-based algorithms (with R and RMSE of 0.385/3.88 km) when ...'.

L608-609: "… notable differences are observed in the CBHs from both ML-based algorithms." Clarify the sentence: Are the differences observed between the two ML-based methods or between the ML-based methods and the observations.

Answer: We have clarified the sentence "... notable differences are observed in the CBHs between both ML-based algorithms...".

L733: Acronym of cloud top temperature.

Answer: Thanks for your suggestion. We have revised it.

L743 the Figure B3b is mentioned but not included in the manuscript or supplements.

Answer: Thanks for your suggestion. Sorry, it is Figure S7b. We have revised it.

Table 1 should be reformatted. Column names should be included. Combining together the common predictors used from satellite measurements for both methods would greatly help make sense of the difference in the input datasets.

Answer: Thanks for your suggestion. We have reformatted Table 1 under your suggestion.

Figure 3: the markers could be bigger (at least in the legend) as the colors are not very distinct between the different methods and appear quite small on the plot.

Answer: Thanks for your suggestion. We use the standard primary colors—red, green, and blue—which offer the highest level of differentiation. Besides, we attempted this, but as the markers grew larger, the dots merged, making it even more difficult to discern. Actually, we modeled this satellite-borne Lidar or CloudRadar profile cross-section comparison by following previous literatures, but it remains difficult to discern. Even when reviewing other papers, a slight zoom-in is often necessary.

Figure 5: In the figure description, it refers to figure 6 and not figure 5.

Answer: Sorry about it. We have revised the figure description for figure 5.

Figure S4: All x-axis legends are misspelling MODIS as "MOIDS".

Answer: Thanks for your suggestion. We have revised this figure.

**Reviewer 2**

The paper provides many insightful results from both physics-based and ML-based algorithms and tremendous observational data sets for satellite cloud base height retrievals. I appreciate the authors' hard work which must include large data set processing. The methods and results are fairly well described in general. However, some major aspects of the analysis and additional information should be provided and clarified, which may lead to different conclusions, depending on the details.

Comments/suggestions:

Introduction: The literature survey on CBH retrievals using satellite observations seems quite narrow, and not very clearly described even though for instance the authors address "two" groups but I cannot find what exactly that means. The paragraphs may be reorganized for further clarification for readers. More detailed comments are below.

Answer: Thanks for your suggestion. We have re-rephrased this paragraph as "*Two groups of retrieval algorithms, one physics-based and the other machine-learning (ML) based, each consisting of two independent approaches, have been developed to retrieve cloud base height (CBH) and its diurnal cycle from Himawari-8 geostationary satellite observations.*" To make it more readable.

For the comparisons of the 4 algorithm outputs: Do the ML algorithms target the CBH of ceilings or still for the topmost layers like physics-based CBH algorithms? I cannot find the details, which might give very different results depending on which is focused on.

In particular, I cannot find sufficient discussions on multilayer cloud cases (how to treat them in each algorithm and how to consider those scenes in comparisons which may give very different analysis results and conclusions).

Answer: Thanks for your suggestion. In this manuscript, the cloud processing is under the assumption that the cloud is a single layer. Due to the relative weak penetration capability of passive geostationary satellite imagers, it is challenging to determine the actual number of cloud layers in vertical during real-time cloud parameter retrieval. Therefore, some classical algorithms (such as NASA/EOS-MODIS (Platnick et al., 2017) and VIIRS (Noh et al ., 2017)) typically assume a single-layer cloud for processing. This assumption is particularly relevant in physical inversion algorithms,

where properties like optical thickness are based on the premise of a single-layer cloud.

To clarify this for the reader, I have added a note "*Note that, similar to previous studies on cloud retrieval (Noh et al. 2017; Platnick et al. 2017), this investigation also assumes a single-layer cloud for all CBH algorithms, due to the challenges associated with determining multilayer cloud structures.*" in Line 278.

Reference

Platnick S, Meyer KG, King MD, Wind G, Amarasinghe N, Marchant B, et al. The MODIS cloud optical and microphysical products: Collection 6 updates and examples from Terra and Aqua. IEEE Trans Geosci Remote Sens 2017, 55(1): 502-525.

Noh Y-J, Forsythe JM, Miller SD, Seaman CJ, Li Y, Heidinger AK, et al. Cloud-base height estimation from VIIRS. Part II: A statistical algorithm based on A-Train satellite data. Journal of Atmospheric and Oceanic Technology 2017, 34(3): 585–598.

Section 4.1: So you use two physics-based CBH methods which are highly dependent on CTH accuracy, right? Then, it would be worth addressing the CTH dependency in the comparison results, which doesn't seem to evaluate cloud top separately here. Additionally, when using 2B-CLDCLASS-LIDAR cloud product, which cloud base data is used here, the topmost layer's one or the lowest? If the lowest from CloudSat is used for the two physics-based cloud base algorithm products which are designed for the uppermost layer cloud base, the comparisons wouldn't be complete, or you should address it.

Answer: Thanks for your suggestion. We have compared our CTH product with MODIS in the Appendix Section A ad the supplementary documentation in Figures S2, S3 and S4. Actually, this is an old product for Himawari-8 and Fengyun-4 GEO satellite. We have cited our previous paper (Min et al., 2020) to explain the accuracy in this manuscript.

We use the lowest cloud base heights from joint CloudSat/CALIPSO cloud product in this study. To highlight this, we have added a sentence "*In this study, we consider the lowest effective cloud base height from the joint CloudSat/CALIOP data*

*as the true values for training and validation.*" in Line 250 to explain this issue. Besides, we also assume the cloud is a single layer as mentioned before.

Reference

Min Min, Jun Li, Fu Wang, Zijing Liu, W. Paul Menzel, 2020. Retrieval of cloud top properties from advanced geostationary satellite imager measurements based on machine learning algorithms [J]. Remote Sensing of Environment, 239: 111616, doi: 10.1016/j.rse.2019.111616

Line 250: For the matchup method, please add references or a brief description including the temporal/spatial matchup windows (here or around line 364).

Answer: Thanks for your suggestion. We have added a sentence and a reference "*In this process, the nearest distance matching method is employed, ensuring that the observation time difference between the CloudSat/CALIPSO observation point and the matched Himwari-8 data is less than 5 minutes (Noh et al. 2017).*" to illustrate this issue.

Line 380: "In their results" - Their comparison was for the topmost layers for both VIIRS and CloudSat, excluding precipitation pixels and only using CTH error is within 2 km range due to CBH's large dependency on CTH. Is the same strategy used here for the comparisons? Otherwise, it should be clearly stated similar to your Fig. 3 discussion below, and if needed, the comparison should be close to apple-to-apple approaches for all four algorithms.

Answer: Thanks for your suggestion. Yes, we used the same comparison method in this study, which is consistent with the description of Noh et al., 2017 for VIIRS vs CloudSat comparison.

Reference

Noh Y-J, Forsythe JM, Miller SD, Seaman CJ, Li Y, Heidinger AK, et al. Cloud-base height estimation from VIIRS. Part II: A statistical algorithm based on A-Train satellite data. Journal of Atmospheric and Oceanic Technology 2017, 34(3): 585–598.

Line 80-81: It seems like a jump in the discussion here. It could be improved with more clarifications and emphasis on the main goal of this study regarding the cloud base diurnal cycle.

Answer: Thank you for your advice. We have move the sentences at Lines 82-93 to make it more clear "*As well known, there are distinct diurnal cycle characteristics of clouds in different regions across the globe (Li et al. 2022). These diurnal cycle characteristics primarily stem from the daily solar energy cycle absorbed by both the atmosphere and Earth's surface. Besides, vertical atmospheric motions are shaped by imbalances in atmospheric heating and surface configurations, also leading to a range of cloud movements and structures (Miller et al. 2018). Cloud base plays a pivotal role in weather and climate processes. It is critical for predicting fog and cloud-related visibility issues important in aviation and weather forecasting. For instance, lower cloud bases often lead to more intense rainfall. In climate modeling, CBH is integral for accurate long-term weather predictions and understanding the radiative balance of the Earth, which influences global temperatures (Zheng and Rosenfeld 2015).*".

Line 100-102: I understand more details will be in the next sessions, but it would be good to add reference papers simply here first for CloudSat, CALIPSO, A-Train, etc.

Answer: Thank you for your advice. We have added references for CloudSat, CALIPSO, A-Train.

**CloudSat**: Stephens, Graeme L., et al. "The CloudSat mission and the A-Train: A new dimension of space-based observations of clouds and precipitation." Bulletin of the American Meteorological Society 83.12 (2002): 1771-1790.

**CALIPSO**: Winker, David M., et al. "Overview of the CALIPSO mission and CALIOP data processing algorithms." Journal of Atmospheric and Oceanic Technology 26.11 (2009): 2310-2323.

**A-Train**: Stephens, Graeme L., et al. "The CloudSat mission and the A-Train: A new dimension of space-based observations of clouds and precipitation." Bulletin of the American Meteorological Society 83.12 (2002): 1771-1790.

Line 114-115

Line 116: "Two primary methods" - out of physics-based methods in passive sensor observations? Which two methods exactly do the authors discuss here, cloud type dependency or CGT and CTH use for CBH? Said two, but cannot find a clear cut for the two groups. If not better, it would be ok to overall write some literature survey to obtain CBH from physics-based (or statistical-based_ methods.

Answer: Thanks for your suggestion. Two primary methods means the method summarized in the first two sentence of Line 130. We have changed this to "*These methods aforementioned are prominent in…*" to make it more readable.

Line 121-126: It looks like the sentences should be divided into two: 1) a general CBH derivation method and 2) Noh et al. which was demonstrated using VIIRS. The first description would be for a general method to derive CBH by subtracting CGT from CTH.

Answer: Thanks for your suggestion. We have re-checked this sentence. We think it only represents the second method using physical theory. Noh and Hutchison et al are the similar method.

Line 128: This algorithm: Which one, Noh et al. or Hutchison et al., or both or the others?

Answer: Thank you for your question. This algorithm is from Hutchison et al. and we have revised it.

Line 148: "root mean square error (RMSE)" No need to use both full name + acronym repeatedly.

Answer: Thank you for your advice. We remained both full name and acronym for the root mean square error (RMSE) for the first time in line 162, as same with correlation coefficient (R). After that, we only used the acronyms RMSE and R.

Line 154-165: As mentioned earlier, it would be better that these discussions on the importance of the diurnal cycle of CBH study are addressed first before jumping into line 80-82 and line 150-153.

Answer: Thank you for your advice. We have moved this part to the Lines 82-93.

Line 284: Add references for CLAVR-x and ACBA.

Answer: Thank you for your advice. We have added references for CLAVR-x and ACBA.

**CLAVR-x**: Noh, Yoo-Jeong, et al. "Cloud-base height estimation from VIIRS. Part II: A statistical algorithm based on A-Train satellite data." Journal of Atmospheric and Oceanic Technology 34.3 (2017): 585-598.

**ACBA**: Noh, Yoo-Jeong, et al. "Enterprise AWG Cloud Base Algorithm (ACBA)." (2022).

Line 288-289: It is recently being applied for NOAA GOES-R ABI as well, as part of the NOAA Enterprise Cloud Algorithms. Correct GORS-R to GOES-R.
Answer: Thank you for your suggestion. We have revised it.

Line 295: This sentence should be corrected such as "This algorithm is suitable for single-layer and the topmost layer of multi-layer clouds"
Answer: Thank you for your nice suggestion. We have revised the sentence.

Line 391-393: Please double check this statement. Do the physics-based CBH algorithms use visible band data? Or you want to say the lack of CloudSat/CALIPSO data below 1 km due to the ground clutter or something? Line 417 statement should be placed first here, too.
Answer: Thank you for your suggestion. We found that this sentence felt a bit out of place in this paragraph, so we decided to omit it.

Line 550-551: Not very correct statement. CLAVR-x CBH uses CWP input from NWP data for nighttime CBH, although the products would be degraded due to the lack of visible band information as the authors mentioned in the paper.
Answer: Thank you for your suggestion. Agree, we have deleted this sentence.

Line 646: Correct it to "Colorado State University".
Answer: Thank you for your suggestion. We have revised it.

---

## Referee Report (RR1)

Technical note: Applicability of physics-based and machine-learning-based algorithms of geostationary satellite in retrieving the diurnal cycle of cloud base height – Wang et al. (egusphere-2024-1516)

This paper addresses several interesting but somewhat diverse research topics, and I appreciate the authors' efforts in conducting this work. However, the manuscript gives the impression that multiple studies are being combined, which can make it difficult to discern whether the primary focus is on the development of the ML algorithms or the accuracy of the CBH products from each algorithm. I believe that the organization and clarity of the presentation could be improved to better highlight the main objectives.

If the focus is on the diurnal cycles of CBHs, some sections, such as the MODIS product evaluation, might not be necessary and may detract from the central message of the paper. A more concise reference to this information, perhaps by adding appropriate citations, could suffice. Streamlining the content in this way could help avoid the level of detail typically found in a graduate thesis and keep the reader's attention on the core contributions of the paper.

Another main concern is, regarding line 251 and others:

I find the lack of discussions or additional explanations of CTH aspects in the CBH eval comparisons, even though those are mentioned in the description part. Probably the evaluation results from Seaman et al. 2017 or Noh et al. 2017 cited in this paper (e.g., line 407- ) were conducted under the "within-spec" condition when CTH is a 2-km error range compared against ground truth data, which aimed to isolate CBH eval, decreasing CTH effects as the physics-based algorithms are highly dependent on CTH accuracy.

No sufficient discussions on multilayers: Multilayer cases (either limitations or future plans) as well as nighttime cases should be addressed in the conclusion.

Detailed comments and questions for further clarification are below.

Comments/questions:

Line 50 "sensor may be attributed to utilizing the same dataset …" : Not clear.

Line 130 "These methods aforementioned are prominent in retrieving CBH … space-based remote" : Not clear, which methods you are referring to?

Line 132 ...”The first method ...”:  It seems like a starting sentence is missing. Clarification needed. What these first and second methods about?

Line 161: “... algorithm, achieving a high correlation coefficient (R) of 0.92 and a low root mean square error (RMSE): add compared against which data?

Line 167-168: It should be partially true, but Tana et al.'s study that the authors cited right above has used Himawari-8 data, which doesn't support this argument. It would be better to replace "mainly" with partially or something similar in line 166 and also better to rewrite line 164-168 to address the diurnal cycles haven't been well investigated in both GEO and LEO remote sensing research.

Line 217: For MODIS data, please add a couple of additional explanations why the authors describe MODIS here with such details (including the MODIS product evaluation in Appendix), in order to help readers' understanding, even though the reason appeared later but not clear yet here in this general Data section. Otherwise, it may distract the main topic of this paper.

Line 272-273: No need to address this here, out of the main focus of this study which adopts only the algorithm for the H8 application, not JPSS VIIRS, anyway.

Line 287: This “reliable” looks already quite deterministic. "Another" should be enough here.

Line 291-292: “... studies have also demonstrated a R of 0.569 and a RMSE of 2.3 km for the JPSS..” -> This is also unnecessary here (no info how the error statistics were obtains won't be helpful to readers, too), slightly out of this study's scope.

Line 311-317: It would be good if this is a thesis, but somewhat too much extra information for the paper. It looks already enough by citing Breiman and Min et al. Tan et al.'s papers.

Line 328: “based on” -> using additional information from NWP model data or similar sentences may be considered. Need to rewrite. It may give the impression that the algorithms rely solely on NWP data.

Line 336-337: but employs different view zenith angles and 337 azimuth angles. -> Not clear. Need more clarification.

Line 338: “matching method” what about parallax corrections between two sensors? It seems some technical details are missing here.

Line 407-412: Did the error statistics consider a similar factor for CTH eval with Seaman et al.'s "within spec" comparisons or just under all cloud conditions?

Line 421: "the CBHs lower than 2 km for"-> Is there any possibility of inversion in the low boundary layers as GFS NWP data may not have such high vertical resolution to resolve and thus CTH errors causing CBH errors in physics algorithms and also NWP input impact on ML algorithms maybe.

Line 477-478: Not necessary to describe all the general lidar observation theory. The paragraph can be trimmed.

Line 493: As the authors addressed, ground lidar observations tend to be quickly attenuated near lowest cloud base especially for thick clouds. If using solely lidar data in comparisons, cloud characteristics (type, depth, etc) related weather conditions would be good to be discussed as well.

Line 527: Line 538 and below details about the radar data should be placed here.

Fig. 1: Additionally, it will be good to mention these comparisons for all cloud conditions including single and multilayer cloud scenes, and something like the CTH accuracy (or evaluation) is not considered.

Figure 2 caption : CBHs -> lowest CBHs, 2017 -> the statistics for all cloud scenes including both single and multilayers? If so, please specify it.

Fig. 3: What happened to the "No CloudSat obs" part on the right end in (b)?

Line 534-538: It's not well organized, which seems like a jump in the context. It should be placed at the beginning of this ground-observation eval section.

Line 648-650: Specify what exactly the factor is.

Line 655: Too early conclusive remark with limited comparisons and without intensive case analyses.

Line 565: "it is more reasonable to opt for physics-based cloud base height algorithms." It seems like a too early conclusive remark with limited comparisons and without intensive case analyses.

Line 670: As well as "nighttime cases", multilayer cases should be mentioned in the conclusion.

Minors:

Line 35: remove 'one'

Line 123: remove "As well known"

Line 134: References in Line 227 should be put here, too, which is the first place for MODIS..

Line 162: the random forest -> RF. The acronym was already defined. Found the same errors in several places.

Line 204: The validation "is" -> has been

Line 335: remove "Global Forecast System" which acronym has been already defined.

Line 375: Not good to use two denotes COT and D_COT in one paper. Please use one consistently.

---

## Author Response (AR2)

**Reviewer 1**

Dear Editor, dear authors,

The authors made an effort to improve the manuscript and to support their presented results. The previously raised points have been properly addressed. I recommend accepting the manuscript for publication after the following few minor/technical points have been addressed:

- L422-428 + L461-472 The presentation of the joint evaluation dataset is still a bit confusing to me. The amounts of samples between the two ML-based methods naturally differ as clearly clarified now. However, the dataset for which the scatter plots of Figure 2 are produced is not clear. I understand the authors compile colocated samples for the year 2017 and then randomly split these samples for training and validation. Is this validation set then used for all the evaluation (plots and metrics) presented? Overall, I would encourage the authors to centralize the information about the joint dataset in the section 3.4 to improve clarity (eg. move the beginning of section 4.1 to this section).

Answer: Thanks for your suggestion. In Figure 2, we used the exact same set of instances to test the four algorithms. As per your suggestion, I moved the beginning part of Section 4.1 to Section 3.4.

- "... physics-based methods ..." (L416)

Answer: Thanks for your nice advice. We have revised the sentence.

- L615 "Figure 5a and 5b..."

Answer: Thanks for your nice advice. We have revised it.

- L741 "... data from 2017 and 2019."

Answer: Thanks for your nice advice. We have revised it.

- Figure 1 "... by the four independent ..."

Answer: Thanks for your nice advice. We have revised it.

- Figure 3 There is one too many (a) indicators in the caption. Using the same colors

for the respective method retrievals as in all the other figures (5, 6, 7, 8) would be perfect.

Answer: Thanks for your nice advice. As per your suggestion, we have revised the Figure 3.

- Figure 7 Remove UTC in the caption as the local time is used.

Answer: Thanks for your nice advice. We have removed UTC.

- Extensive review of the use of acronyms was done by the authors in the revised manuscript, just very few cases remain (L326, L392, L666 in track changes).

Answer: Thanks for your nice advice. We have unified the description of all cloud base height in this paper.

**Reviewer 2**

Technical note: Applicability of physics-based and machine-learning-based algorithms of geostationary satellite in retrieving the diurnal cycle of cloud base height – Wang et al. (egusphere-2024-1516)

This paper addresses several interesting but somewhat diverse research topics, and I appreciate the authors' efforts in conducting this work. However, the manuscript gives the impression that multiple studies are being combined, which can make it difficult to discern whether the primary focus is on the development of the ML algorithms or the accuracy of the CBH products from each algorithm. I believe that the organization and clarity of the presentation could be improved to better highlight the main objectives.

If the focus is on the diurnal cycles of CBHs, some sections, such as the MODIS product evaluation, might not be necessary and may detract from the central message of the paper. A more concise reference to this information, perhaps by adding appropriate citations, could suffice. Streamlining the content in this way could help avoid the level of detail typically found in a graduate thesis and keep the reader's attention on the core contributions of the paper.

Another main concern is, regarding line 251 and others:

I find the lack of discussions or additional explanations of CTH aspects in the CBH eval comparisons, even though those are mentioned in the description part. Probably the evaluation results from Seaman et al. 2017 or Noh et al. 2017 cited in this paper (e.g., line 407- ) were conducted under the "within-spec" condition when CTH is a 2-km error range compared against ground truth data, which aimed to isolate CBH eval, decreasing CTH effects as the physics-based algorithms are highly dependent on CTH accuracy.

No sufficient discussions on multilayers: Multilayer cases (either limitations or future plans) as well as nighttime cases should be addressed in the conclusion.

Detailed comments and questions for further clarification are below.

Answer: Thanks for your nice advice. We believe your suggestion is absolutely right; indeed, some parts of the text were overly verbose, making it hard for readers to grasp the key points. We have already made some revisions, such as cutting down unnecessary wording and rearranging sections, like moving Section 4.1 to Section 3.4, removing some descriptions in section 3.3.

Appendix A section has discussed and validated the accuracies of cloud products from H8 satellite. We have provided a detailed introduction in the second paragraph of Chapter 2 (Data) and made a thorough comparison in Appendix A. In fact, several early papers have already conducted detailed comparisons. In Section 4.1, we once again cited the paper by Min et al., 2020, to describe the accuracy of CTH used in this study *"According to previous CALIPSO validations (Min et al., 2020), the absolute bias of cloud top height retrieved by the H8 satellite is approximately 3 km, with an absolute bias of 1 to 2 km for samples below 5 km. The accuracy of CTH is crucial for estimating CBH in the subsequent algorithm."*.

Moreover, since it is not easy to exactly distinguish between multilayer and single-layer clouds solely based on geostationary satellites passive sensor, neither the early physical algorithms from NOAA and NASA nor the newly developed ML-based methods are effective at making this distinction (Baker, 2011; Noh et al., 2017). It assumes the single layer for cloud base heigh retrieval of all samples. The algorithm will retrieve primarily the cloud base of the uppermost layer cloud if lower clouds are

not well detected and the column-integrated cloud water path (CWP) retrieval is made for the topmost layer, which shows that the accuracy and representativeness of the upstream CWP retrieval is essential to the CBH estimate together with the CTH accuracy. It should be noted that multilayer cloud scenes are typically more challenging for CTH and CBH retrievals, and CLAVR-x CTH also often tends to be biased low for the multilayer situation (Noh et al., 2017). Therefore, we did not classify the clouds into multilayer and single-layer categories in this study. However, we have added some sentence in our manuscript to underscore this issue at the beginning of Section 3.1 "*It is important to note that multilayer cloud scenes remain a challenge for retrieving both CTH and CBH, especially when considering the column-integrated cloud water path (CWP) used in physics-based algorithms (Noh et al. 2017). In this study, we will simplify the scenario by assuming a single-layer cloud for all algorithms.*" .

Reference

Min Min, Jun Li*, Fu Wang, Zijing Liu, W. Paul Menzel, 2020. Retrieval of cloud top properties from advanced geostationary satellite imager measurements based on machine learning algorithms [J]. Remote Sensing of Environment, 239: 111616, doi: 10.1016/j.rse.2019.111616

Baker, N.: Joint Polar Satellite System (JPSS) VIIRS Cloud Base Height Algorithm Theoretical Basis Document (ATBD), 2011.

Noh, Y.-J., Forsythe, J. M., Miller, S. D., Seaman, C. J., Li, Y., Heidinger, A. K., Lindsey, D. T., Rogers, M. A., and Partain, P. T.: Cloud-base height estimation from VIIRS. Part II: A statistical algorithm based on A-Train satellite data, Journal of Atmospheric and Oceanic Technology, 34, 585–598, 10.1175/JTECH-D-16-0110.1, 2017.

Comments/questions:

Line 50 "sensor may be attributed to utilizing the same dataset …" : Not clear.

Answer: Thanks for your nice advice. We have added "…of CloudSat/CALIOP… " to make it more clear at this Line.

Line 130 "These methods aforementioned are prominent in retrieving CBH … space-based remote" : Not clear, which methods you are referring to?

Answer: Thanks for your nice advice. We have revised it as "*These passive space-based remote sensing methods aforementioned, such as satellite imagery, play a key role in retrieving CBH.*" to make it more clear at this Line.

Line 132 …"The first method …": It seems like a starting sentence is missing. Clarification needed. What these first and second methods about?

Answer: Thanks for your nice advice. We have revised it as "*In terms of detection principles, the first method…*" to make it more clear at this Line.

Line 161: "… algorithm, achieving a high correlation coefficient (R) of 0.92 and a low root mean square error (RMSE): add compared against which data?

Answer: Thanks for your nice advice. We have revised it as "*…compared with CloudSat/CALISPO data.*" to make it more clear at Line 164.

Line 167-168: It should be partially true, but Tana et al.'s study that the authors cited right above has used Himawari-8 data, which doesn't support this argument. It would be better to replace "mainly" with partially or something similar in line 166 and also better to rewrite

Answer: Thanks for your nice advice. We have revised it as "…. achieving a similar high.." at line 162.

line 164-168 to address the diurnal cycles haven't been well investigated in both GEO and LEO remote sensing research.

Answer: Thanks for your nice advice. Agree, we have added a sentence of "*The diurnal cycles of CBH have not been well investigated in both GEO and LEO remote sensing research.*" at lines 169-170.

Line 217: For MODIS data, please add a couple of additional explanations why the authors describe MODIS here with such details (including the MODIS product evaluation in Appendix), in order to help readers' understanding, even though the

reason appeared later but not clear yet here in this general Data section. Otherwise, it may distract the main topic of this paper.

Answer: Thanks for your nice advice. We have added a sentence of "*High-quality, long-term series MODIS data is often used as a validation reference to evaluate the products of new satellites.*" to make it more clear at lines 226-227.

Line 272-273: No need to address this here, out of the main focus of this study which adopts only the algorithm for the H8 application, not JPSS VIIRS, anyway.

Answer: Thanks for your nice advice. Agree, we have deleted this sentence.

Line 287: This "reliable" looks already quite deterministic. "Another" should be enough here.

Answer: Thanks for your nice advice. Agree, we have revised it.

Line 291-292: "… studies have also demonstrated a R of 0.569 and a RMSE of 2.3 km for the JPSS.. " -> This is also unnecessary here (no info how the error statistics were obtains won't be helpful to readers, too), slightly out of this study's scope.

Answer: Thanks for your nice advice. I still hope to keep this sentence, as it will help readers compare the accuracy of our work with previous studies.

Line 311-317: It would be good if this is a thesis, but somewhat too much extra information for the paper. It looks already enough by citing Breiman and Min et al. Tan et al.'s papers.

Answer: Thanks for your nice advice. We have deleted some descriptions about this part.

Line 328: "based on"-> using additional information from NWP model data or similar sentences may be considered. Need to rewrite. It may give the impression that the algorithms rely solely on NWP data.

Answer: Thanks for your nice advice. Agree, we have revised many "based on.." in our manuscript.

Line 336-337: but employs different view zenith angles and azimuth angles. -> Not clear. Need more clarification.

Answer: Thanks for your nice advice. IR-single model only uses view zenith and azimuth angles (no solar zenith and azimuth angles). Hence, we have revised it as "... *but employs only view zenith angles and azimuth angles.*" at line 334.

Line 338: "matching method" what about parallax corrections between two sensors? It seems some technical details are missing here.

Answer: Thanks for your nice advice. We think there is no parallax correction between two sensors. It only means that the nearest distance matching method is used to match two different product in space and time domains. We have revised this sentence at line 384 "... *ensuring that collocating the closest points and the observation time difference between the CloudSat/CALIPSO observation point and the matched Himwari-8 data is less than 5 minutes.*" to further explain this "matching method" here.

Line 407-412: Did the error statistics consider a similar factor for CTH eval with Seaman et al.'s "within spec" comparisons or just under all cloud conditions?

Answer: Thanks for your question. We use the similar factors for CBH evaluation as Seaman et al., 2017. We have described this at the end of section 4.1.1.

Line 421: "the CBHs lower than 2 km for"-> Is there any possibility of inversion in the low boundary layers as GFS NWP data may not have such high vertical resolution to resolve and thus CTH errors causing CBH errors in physics algorithms and also NWP input impact on ML algorithms maybe.

Answer: Thanks for your nice advice. The reason you mentioned is possible, but based on the inversion results of the cloud top height, it doesn't seem very likely. The main reason is probably that the lower cloud base signals are difficult to detect from the satellite using infrared or visible light channels.

Line 477-478: Not necessary to describe all the general lidar observation theory. The paragraph can be trimmed.

Answer: Thanks for your nice advice. We have deleted some sentences in this paragraph.

Line 493: As the authors addressed, ground lidar observations tend to be quickly attenuated near lowest cloud base especially for thick clouds. If using solely lidar data in comparisons, cloud characteristics (type, depth, etc) related weather conditions would be good to be discussed as well.

Answer: Thanks for your nice advice. We have added a sentence of "*These two days have been cloudy, with stratiform clouds at an altitude of around 5 km and no precipitation occurring.*" to make it more clear.

Line 527: Line 538 and below details about the radar data should be placed here.

Answer: Thanks for your nice advice. Agree, we have moved this paragraph forward as your suggestion.

Fig. 1: Additionally, it will be good to mention these comparisons for all cloud conditions including single and multilayer cloud scenes, and something like the CTH accuracy (or evaluation) is not considered.

Answer: Thanks for your nice advice. Agree, we have added a sentence "*...RF IR-single algorithm for all cloud conditions including single and multilayer cloud scenes.*" at Line 391 to further explain this issue. We have compared CTH in our supplementary documentation. Also we have explained the accuracy of CTH in Section 4.1 mentioned before.

Figure 2 caption : CBHs -> lowest CBHs, 2017 -> the statistics for all cloud scenes including both single and multilayers? If so, please specify it.

Answer: Thanks for your nice advice. Yes, the statistics are for all cloud scenes including both single and multilayer clouds. As we mentioned earlier, passive remote sensing satellites have difficulty distinguishing between single-layer and multi-layer clouds, especially at night. We have added an interpretation in this caption.

Fig. 3: What happened to the "No CloudSat obs" part on the right end in (b)?

Answer: Thanks for your nice advice. We believe there are clouds, just not clouds with very strong Radar echoes.

Line 534-538: It's not well organized, which seems like a jump in the context. It should be placed at the beginning of this ground-observation eval section.

Answer: Thanks for your nice advice. Agree, we have moved this paragraph forward to make it more clear.

Line 648-650: Specify what exactly the factor is.

Answer: Thanks for your nice advice. We have added a sentence "*Therefore, this factor induced by detection principle could contribute…*" at line 644 to make it more clear.

Line 655: Too early conclusive remark with limited comparisons and without intensive case analyses.

Answer: Thanks for your nice advice. We have changed this sentence as "*Ideally, we guess that including more spaceborne cloud profiling radars with varying passing times (covering the entire day) in the training dataset could improve the machine learning technique, potentially leading to a higher-quality CBH product with more comprehensive observations.*".

Line 565: "it is more reasonable to opt for physics-based cloud base height algorithms. " It seems like a too early conclusive remark with limited comparisons and without intensive case analyses.

Answer: Thanks for your nice advice. Agree, we have deleted this sentence.

Line 670: As well as "nighttime cases" , multilayer cases should be mentioned in the conclusion.

Answer: Thanks for your nice advice. We have added a sentence "*… CBHs from GEO H8/AHI data under the assumption of single layer cloud.*" at the second paragraph to explain this issue.

Minors:
Line 35: remove 'one'
Answer: Thanks for your nice advice. We have removed it.

Line 123: remove "As well known"
Answer: Thanks for your nice advice. We have removed it.

Line 134: References in Line 227 should be put here, too, which is the first place for MODIS.

Answer: Thanks for your nice advice. We have added the references in line 136.

Line 162: the random forest -> RF . The acronym was already defined. Found the same errors in several places.

Answer: Thanks for your nice advice. We have checked again and revised the corresponding words in the entire text.

Line 204: The validation "is"-> has been

Answer: Thanks for your nice advice. We have revised it.

Line 335: remove "Global Forecast System" which acronym has been already defined.

Answer: Thanks for your nice advice. We have removed it.

Line 375: Not good to use two denotes COT and D_COT in one paper. Please use one consistently.

Answer: Thanks for your suggestion. We replaced COT with $D_{COT}$ have used one denote $D_{COT}$ consistently.

---

## Author Response (AR3)

**Public justification (visible to the public if the article is accepted and published)**:

Dear authors

Thank you for addressing the reviewer comments in this latest round of revisions. I went through the manuscript again and have one minor comment and a few typographical suggestions left.

Comment: The main motivation of the manuscript is to "assess the applicability of physics-based and ML-based algorithms in retrieving the diurnal cycle of cloud base height", and this is discussed in Sec. 4.2.2. However, in the conclusions & discussion section, the performance of the algorithms in retrieving the diurnal cycle of CBH is completely left out of the discussion - only the general performance is discussed. I thus ask the authors to rewrite this section to emphasise the diurnal cycle aspect of this technical note.

Typographical suggestions (line numbers refer to the track-changes doc):

L83: remove 'As well known' and start a new paragraph.

Answer: Thanks for your suggestion. Yes, we have removed them and started a new paragraph.

L133, L139, L344, L458, L506, L510, L514 : in-text (rather than in-brackets) citation, e.g. Yang et al. (2021). Check out the entire manuscript again if the citations are correctly formatted.

Answer: Thanks for your suggestion. We have checked out the entire manuscript again and corrected the relevant citation formats.

L164: properly cite ERA5 and not just give the URL.

Answer: Thanks for your suggestion. We have cited a reference about ERA5 (Lin et al., 2022) in line 157.

L280: write out Physics and machine-learning based … (instead of '/')

Answer: Thanks for your nice advice. We have revised it in line 262.

L329: this sentence needs a reference

Answer: Thanks for your nice advice. We have added the reference of Baker, 2011.

L460ff: The R and other values should be rounded to two digits, not three.

Answer: Thanks for your nice advice. We have revised relevant values to two digits in the entire manuscript.

L547: …emits laser pulses …

Answer: Thanks for your nice advice. We have revised it.

L555: … lidar return signal of cloud droplets is markedly distinct..

Answer: Thanks for your nice advice. We have revised it.

L623: remove 'as well known', add "over land" in: diurnal variation of cloud base height over land …

Answer: Thanks for your nice advice. We have revised as your suggestion in line 534.

L630: adjust the transition from the shifted block (in green) to "Therefore, it is essential…". This doesn't flow as it currently is in the manuscript.

Answer: Thanks for your nice advice. The colors of the words in this paper are black.

---

## Author Response (AR4)

Dear authors

Thank you for applying the typographical suggestions in this latest round of revisions. Unfortunately, I don't see my main comment from the last round addressed. I therefore ask you again to reply to this comment from the last round.

Comment: The main motivation of the manuscript is to "assess the applicability of physics-based and ML-based algorithms in retrieving the diurnal cycle of cloud base height", and this is discussed in Sec. 4.2.2. However, in the conclusions & discussion section, the performance of the algorithms in retrieving the diurnal cycle of CBH is completely left out of the discussion - only the general performance is discussed. I thus ask the authors to rewrite this section to emphasise the diurnal cycle aspect of this technical note.

For the reference to ERA5, please also refer to the original publication of Hersbach et al. 2020 (https://doi.org/10.1002/qj.3803) to acknowledge the work.

Once these comments are properly addressed, I think the manuscript could be published.

Best wishes, Raphaela Vogel

Answer: Thanks for your suggestion. Yes, agree, we have added a paragraph at the discussion section to illustrate this issue "*In general, the physics-based algorithms, such as GEO CLAVR-x and GEO IDPS, demonstrate notable advantages in capturing the diurnal cycle of CBH. Unlike ML-based methods, they offer more stable error metrics, especially with higher correlation and lower RMSE during the daytime. Additionally, they are more effective at capturing significant and natural variations in CBH, providing generally higher quality retrievals from H8/AHI data, even though challenges remain in accurately retrieving CBHs below 1 km.*" at line 638.

Besides, we have added Hersbach et al. 2020 (https://doi.org/10.1002/qj.3803) paper for the reference to ERA5 at line 157. But we did not use ERA5 data in this work. ERA5 is only used in the previous work (Lin et al., 2022), and the main description is around line 157

Lin, H., Li, Z., Li, J., Zhang, F., Min, M., and Menzel, W. P.: Estimate of daytime single-layer cloud base height from Advanced Baseline Imager measurements, Remote Sensing of Environment, 274, 112970, 10.1016/j.rse.2022.112970, 2022.